# FairlyUncertain: A Comprehensive Benchmark of Uncertainty in Algorithmic Fairness

**Lucas Rosenblatt**[*] **& R. Teal Witter**[*]
New York University
{lucas.rosenblatt,rtealwitter}@nyu.edu

## Abstract

Fair predictive algorithms hinge on both equality and trust, yet inherent uncertainty in real-world data challenges our ability to make consistent, fair, and calibrated decisions. While fairly managing predictive *error* has been extensively explored, some recent work has begun to address the challenge of fairly accounting for irreducible prediction *uncertainty*. However, a clear taxonomy and well-specified objectives for integrating uncertainty into fairness remains undefined. We address this gap by introducing `FairlyUncertain`, an axiomatic benchmark for evaluating uncertainty estimates in fairness. Our benchmark posits that fair predictive uncertainty estimates should be *consistent* across learning pipelines and *calibrated* to observed randomness. Through extensive experiments on ten popular fairness datasets, our evaluation reveals: (1) A theoretically justified and simple method for estimating uncertainty in binary settings is more consistent and calibrated than prior work; (2) Abstaining from binary predictions, even with improved uncertainty estimates, reduces error but does not alleviate outcome imbalances between demographic groups; (3) Incorporating consistent and calibrated uncertainty estimates in regression tasks improves fairness without any explicit fairness interventions. Additionally, our benchmark package is designed to be extensible and open-source, to grow with the field. By providing a standardized framework for assessing the interplay between uncertainty and fairness, `FairlyUncertain` paves the way for more equitable and trustworthy machine learning practices.

## 1 Introduction

Fairness in machine learning enhances transparency and trust in algorithmic predictions, and is both a legal and moral imperative given the direct impact predictive models can have on peoples' lives. Although extensive research has addressed reducing predictive error disparities among demographic groups – by tackling limited data, model biases, or structural inequities – achieving fairness also necessitates accurately assessing the uncertainty associated with each prediction; however, the precise interplay between algorithmic fairness and uncertainty estimates remains an open question (Bhatt et al., 2021; Hendrickx et al., 2024).

Assessing the interplay between fairness and uncertainty is challenging due to a lack of principled objectives for integrating the two notions. Much work has contended with how to define and measure fairness in predictive models (Lee, 2018; Caton & Haas, 2024; Mitchell et al., 2021; Barocas et al., 2023; Corbett-Davies et al., 2023). Additionally, predictions made on real-world data have inherent *uncertainty*, given factors like noisy individual behavior, measurement errors, and environmental influences. In statistics and machine learning, a careful typology helps organize these sources of randomness. Uncertainty or variance that is specific to each observation is said to display *heteroscedasticity* (as opposed to *homoscedasticity*, which is not a function of the observation itself). Heteroscedastic uncertainty can either be *epistemic* or *aleatoric* in nature (Hüllermeier & Waegeman, 2021). Epistemic uncertainty is uncertainty that can be reduced with more data (Chen et al., 2018); while aleatoric uncertainty is uncertainty that arises from the inherent data distribution, and cannot be reduced with larger samples. We will focus on models that provide a single uncertainty estimate for

---

[*]Truly equal contribution, in alphabetical order

each observation-specific prediction, but will carefully reason about the *aleatoric* component of that uncertainty estimate, with the goal that this quantity is *consistent with* and *calibrated to* the data.

As a concrete example, consider the task of predicting student performance on a standardized exam: one student reliably earns the same score on the test while another student, perhaps because of their home situation, earns a significantly different score depending on unobservable factors in their daily life. In Figure 1, we hypothesize about how differing unseen environmental factors might affect outcome variance. We can produce more accurate *epistemic* uncertainty estimates by incorporating more data and improving our predictive model. However, if we cannot have the same student take the test multiple times, we cannot precisely estimate *aleatoric* uncertainty because we lack repeated observations of a student's performance measuring individual fluctuations under identical conditions. We provide a typology over uncertainty in predictive modeling in Table 5.

Its worth noting that estimating forms of heteroscedastic uncertainty – uncertainty at the individual level – in the context of algorithmic fairness has received substantial recent interest (Liu et al., 2022; Ali et al., 2021; Han et al., 2022; Tahir et al., 2023; Wang et al., 2024b). Some approaches use ensembles of models to estimate uncertainty while others train models to learn the uncertainty directly. In both cases, the models are trained through a machine learning pipeline on a particular architecture with specific hyperparameters. Unfortunately, the estimates can vary substantially with respect to a model's depth, activation function, and other settings (Lakshminarayanan et al., 2017; Guo et al., 2017; Malinin & Gales, 2018). In the next section, we specify *consistency* and *calibration* as *axiomatic principles* for successful and fair uncertainty estimates; these principles guide our construction of the `FairlyUncertain` benchmark.

## 2 PRELIMINARIES

In Section 2.1, we state guiding axioms for *consistent* and *calibrated* uncertainty estimates, as evaluated by the `FairlyUncertain` benchmark. First, however, we must define a *learning pipeline* (given in Definition 2.1, related to definitions from (Black et al., 2022a; Long et al., 2024)), and what makes two learning pipelines *similar* (given in Definition 2.2).

**Definition 2.1** (Learning Pipeline). *Let $\mathcal{T}$ be a training set of $n$ observations with covariates $\mathbf{x}_1, \ldots, \mathbf{x}_n \in \mathcal{X}$, protected attributes $a_1, \ldots, a_n \in \mathcal{A}$, and outcomes $y_1, \ldots, y_n \in \mathcal{Y}$. Consider a set of $m$ hyperparameters $\lambda_1, \ldots, \lambda_m$. A learning pipeline $\mathcal{P}$ is a randomized training procedure that takes the training set and hyperparameters to learn a predictive function $f : \mathcal{X} \times \mathcal{A} \rightarrow \mathcal{Y} \times \mathbb{R}$. The predictive function produces an estimate of the outcome $\hat{y}$ and uncertainty $\sigma \geq 0$ for given covariates $\mathbf{x}$ and protected attributes $a$.*

**Definition 2.2** (Similar Learning Pipelines). *For each hyperparameter $\lambda_j$, let $\tau_k$ represent a threshold that takes on a reasonable value with respect to hyperparameter $j$. Two learning pipelines $\mathcal{P}$ and $\mathcal{P}'$ (Definition 2.1) are considered similar if they differ only in hyperparameter settings i.e., there is some $j \in [m]$ for which $\lambda_k = \lambda'_k$ except when $j = k$ and $|\lambda_j - \lambda'_j| \leq \tau_j$.*

### 2.1 AXIOMS

The `FairlyUncertain` benchmark focuses on two properties that uncertainty estimates produced by *similar learning pipelines* (Definitions 2.1 and 2.2) should satisfy. Let $f_P$ be a predictive function from a learning pipeline $P$ such that $f_P : (\mathbf{x}, a) \mapsto (\mu, \sigma)$ i.e. $f_P$ maps inputs $\mathbf{x}$ and protected

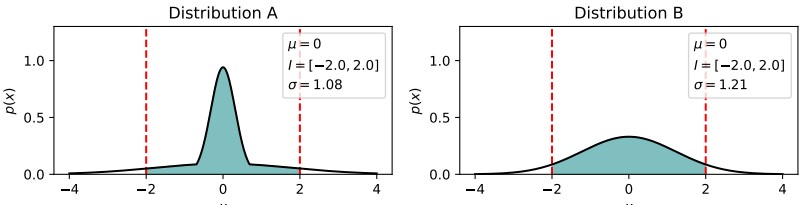

Figure 1: Two distributions over observable outcomes. For example, Distribution A can represent the test scores of a student in a stable home whereas Distribution B can represent the test scores of a student in an unstable home. While both have the same mean and $80\%$ confidence interval, the distributions are substantially different as captured by the standard deviation.

attributes $a$ to a point prediction $\mu$ and an estimated standard deviation of the inherent **(A)** and **(B)** types of uncertainty $\sigma$ for the predicted value of $\mu$. Note if $f_P$ is a classification model, $\mu$ is a probability, and if $f_P$ is a regression model, $\mu$ is a real value.

Now, we are ready to state Axiom 2.3, which formalizes the idea that uncertainty estimates should be a function of the data rather than of an arbitrary learning pipeline. In other words, *similar learning pipelines applied to the same dataset should produce similar uncertainty estimates.*

**Axiom 2.3** (Consistency). *For $f$ and $f'$, which are produced by similar learning pipelines, $\sigma$ and $\sigma'$ should be close. This means that if two learning pipelines are similar, the uncertainty estimates they produce should not vary much.*

This idea is related to the motivation behind selective ensemble and self-consistency ensemble methods (Black et al., 2022a; Cooper et al., 2024). However, Axiom 2.3 is a property of uncertainty estimates rather than outcome predictions.

Consistency by itself is not sufficient; a pipeline that always leads to the same predictions is consistent but not meaningful. This leads to our second property, which posits that uncertainty estimates should also be calibrated. In other words, *uncertainty estimates should always explain observed heteroscedastic variance.*

**Axiom 2.4** (Calibration). *The predicted uncertainty estimates should satisfy $Var(y \mid \mathbf{x}, a) = \sigma^2$, where $\sigma$ is the estimated standard deviation of the outcome $y$ given covariates $\mathbf{x}$ and protected attributes $a$. This means the predicted uncertainty $\sigma$ should match the actual variability observed in the data, capturing the heteroscedastic nature of the uncertainty. When uncertainty estimates are biased or incorrect on average, they fail to be calibrated.*

**Connecting Consistency and Calibration to Fairness Principles.** Uncertainty estimation is widely recognized as a crucial aspect of transparent machine learning practices (Bhatt et al., 2021; Hendrickx et al., 2024). But how is it connected to the concept of fairness? Uncertainty arises from variability in training data and randomness in learning algorithms, leading to a distribution of possible models rather than a single deterministic one. Ignoring this distribution risks making arbitrary decisions, especially for individuals whose predictions might vary across modeling decisions or other sources of uncertainty. Such arbitrariness could disproportionately and unfairly affect minority groups in data Tahir et al. (2023).

Recent work has demonstrated that state-of-the-art fairness interventions can exacerbate *predictive arbitrariness*; models with similar fairness and accuracy performance but different parameters can assign vastly different predictions to individuals, and this arbitrariness is intensified by fairness constraints (Long et al., 2024; Cooper et al., 2024). Our axiom of *consistency* for fair uncertainty estimation builds upon this insight by asserting that uncertainty estimates should not vary significantly across similar learning pipelines. Furthermore, our axiom of *calibration* aims to prevent systematic biases in uncertainty estimates that could disadvantage certain groups. For instance, if uncertainty is consistently underestimated for a particular group, the model may overstate its confidence in predictions for that group, leading to unfair treatment (Ali et al., 2021). This leads us to argue that adhering to the axioms of *consistency* and *calibration* are necessary tenets of a fair uncertainty estimation process.

Once uncertainty estimates are *consistent* and *calibrated*[1], the next challenge is figuring out how to integrate them into fair algorithms effectively. Integrating uncertainty in social settings is commonly done through *abstention*; for a sample where the level of uncertainty is too high, defer making a prediction, thereby avoiding unreliable results (Black et al., 2022a; Cooper et al., 2024). Another method (commonly adopted in regression settings) is to return the uncertainty estimates alongside the predictions, providing a clearer picture of the confidence in the results and allowing practitioners to make more informed decisions (Liu et al., 2022; Wang et al., 2024b). In either setting, no consensus best approach for incorporating uncertainty estimates into fairness interventions has emerged; this motivates our benchmark, `FairlyUncertain`.

---

[1]We use *consistent* and *calibrated* to denote the formal/axiomatic definitions throughout, as opposed to common usage. Additionally, we note that Axioms 2.3 and 2.4 contain specific assumptions that dictate strategies for evaluation; different formalizations would lead to different strategies. We acknowledge this limitation, but argue that any alternative strategies for evaluating uncertainty should intuitively align with each desiderata.

## 3   FAIRLYUNCERTAIN: A BENCHMARK FOR UNCERTAINTY IN FAIRNESS

**Contributions**   To the best of our knowledge, `FairlyUncertain` is a first of its kind fairness *and* uncertainty benchmark. Prior work has either provided singular uncertainty evaluations (Cooper et al., 2024) or extensive fairness benchmarking *without* methods to incorporate uncertainty (Bird et al., 2020; Bellamy et al., 2018; Cruz et al., 2022). `FairlyUncertain` fills the gap, introducing a variety of tests and experiments, and is designed to be extensible and grow with the literature. `FairlyUncertain` supports five binary datasets and five regression datasets off-the-shelf that are standard to the fair machine learning context; Table 6 (deferred to the Appendix) provides prediction task summaries for each. All of our code for `FairlyUncertain` is ready for release as a Python package (available in the supplementary material); the benchmark is modular, so that uncertainty methods and datasets can be easily added. The package includes functions to generate all of our experiments with just a few lines of code (see Appendix C).

**Road Map**   In Section 4, we begin by evaluating *consistency* and *calibration* on both classification and regression tasks. We find that baseline ensemble methods are the most *consistent*. Additionally, we find that in classification tasks, complex ensemble-based methods from prior work on uncertainty are neither *consistent* nor *calibrated*. In contrast, methods that directly learn uncertainty parameters via negative log-likelihood are both *consistent* and *calibrated*. Next, we incorporate uncertainty estimates into making fair decisions. In Section 5, we show that while abstaining from predictions can improve accuracy, this approach does not reduce the imbalance in outcomes between demographic groups. In Section 6, our results lead us to advocate for outputting both predictions and uncertainty estimates in regression tasks. To facilitate this, we introduce *Uncertainty-Aware Statistical Parity* (Definition 6.2), a natural generalization of statistical parity in the regression setting that includes uncertainty estimates. Remarkably, we find that *consistent* and *calibrated* uncertainty methods can reduce *Uncertainty-Aware Statistical Parity* without *any* explicit fairness interventions.

## 4   CONSISTENCY AND CALIBRATION

In this section, we evaluate methods for capturing heteroscedastic uncertainty according to Axioms 2.3 and 2.4. `FairlyUncertain` provides an evaluation strategy for *consistency* that investigates the sensitivity of heteroscedastic uncertainty estimates to hyperparameters in the learning pipeline (Figure 2). For learning pipelines that vary only in one hyperparameter, we compute uncertainty estimates and then the *standard deviation* of these estimates (Table 1).

Measuring *calibration* is much harder since we cannot observe more samples from the same distribution (type **(A)** uncertainty) and would need generated data for type **(B)**. Thus, we devise strategies which measure uncertainty types **(C)**, **(D)** and **(E)** to *approximately* test *calibration* in a **qualitative** and **quantitative** manner. **Qualitatively**, `FairlyUncertain` measures *calibration* over *similar groups,* based on the following intuition: if estimated uncertainties are *calibrated*, observations with higher estimated uncertainties should have higher variation in the test data. Concretely, we create groups with similar uncertainty estimates; then, within each group, we compute the empirical standard deviation of the residual difference between true and predicted outcomes (Figure 3).

Now, imagine we have access to the true probability $p_i$ that individual $i$ will belong to the positive class in the binary setting. Then, following the binomial distribution, the true uncertainty as measured by variance would be $\sigma^2 = p_i(1 - p_i)$. With our **quantitative** strategy for measuring *calibration*, we contend that if we have a prediction for the true probability $\tilde{p}_i \approx p_i$ then a natural prediction for the uncertainty as measured by variance is $\tilde{\sigma}^2 = \tilde{p}_i(1 - \tilde{p}_i)$. A canonical method to evaluate the goodness of fit given probabilistic assumptions is *Negative Log Likelihood* (NLL) Fisher (1925).

This leads to the quantitative evaluation given in Table 1, where we make this assumption on the *meaning* of uncertainty estimates in binary predictions to offer a clear and simple measure of *calibration* based on the NLL of observing the data according to the estimated parameters of the Binomial distribution, interpreting estimate $\mu$ as Binomial probability $p$ (with uncertainty estimate $\sigma$ being the standard variance $p(1 - p)$). Because there are existing methods which produce uncertainty estimates that cannot be interpreted as standard deviations (namely Black et al. (2022a); Cooper et al. (2024)), when assessing these methods, one should focus on the output of the **qualitative** assessment (Figure 3), which provides a more general purpose approach for approximately testing *calibration*.

## 4.1 MODELS AND CLASSIFICATION TASKS

We subject several methods for heteroscedastic uncertainty estimation on binary classification tasks to our evaluation. Many of the methods we consider rely on an ensemble of $k$ models, where each model in the ensemble is trained on samples taken randomly with replacement from the training set. For predictions, these methods simply output the mode prediction of the $k$ models, while for uncertainty estimates, they differ in their calculations. We'll let $k^{(0)}$ be the number of models predicting 0 and $k^{(1)}$ be the number of models predicting 1. Our experiments are run with the following methods:

- The *Ensemble* computes the standard deviation of the $k$ predictions to estimate uncertainty.
- The *Selective Ensemble* method of Black et al. (2022a) estimates uncertainty by computing the $p$-value of observing the number of negative predictions $k^{(0)}$ and the number of positive predictions $k^{(1)}$ were they sampled from a binomial distribution with probability $\frac{1}{2}$.
- The *Self-consistency Ensemble* method of Cooper et al. (2024) estimates uncertainty by computing a so-called self-consistency metric $1 - 2k^{(0)}k^{(1)}/(k(k-1))$. Since the self-consistency metric increases with certainty, we will report self-(in)consistency, i.e., $2k^{(0)}k^{(1)}/(k(k-1))$ for parity with the other approaches.
- The *Binomial NLL* method contrasts with the ensemble methods by directly estimating uncertainties under the Binomial assumption e.g. by producing probabilities $p$ that minimize negative log likelihood on the training set, yielding an uncertainty estimate that is the Binomial standard deviation $\sigma = \sqrt{p(1-p)}$. We offer a brief formal justification as to why we expect *Binomial NLL* to produce good uncertainty estimates in the binary classification setting in Appendix Section D.

**Consistency** Our first experiment evaluates consistency between similar learning pipelines. We do this by varying a hyperparameter setting for our model class; ideally, we would select parameters that affect how the model makes predictions, but shouldn't substantially affect the error rate of those predictions. We note that `FairlyUncertain` makes it easy to experiment with many hyperparameter settings. In this paper, we first experiment with the XGBoost hyperparameter *max_depth*; *depth* is widely applicable to many kinds of models and there are often many equally-valid settings (`FairlyUncertain` ensures that all depths produce models with similar accuracy). The second is *reduction_threshold* $\gamma$, which is specific to the XGBoost model, and smoothly interpolates between encouraging more or less complex models through tree splits. Figure 2 plots heteroscedastic uncertainty estimates for all individuals in each dataset for *max_depth* (see Figure 8 for the *reduction_threshold* plot, and see Figure 12 for a more granular view where we plot the consistency ranges for *each individual* in the data, both in the Appendix). For some individuals, all the methods consistently output the same uncertainty estimates but, for others, estimates produced by the *Selective Ensemble* and *Self-(in)consistency Ensemble* vary drastically. We measure the overall *consistency* of each method by computing the empirical standard deviation of the estimates at each depth. Table 1 reports the maximum individual empirical standard deviation for each algorithm and dataset. The *Ensemble* algorithm is the most consistent, followed closely by *Binomial NLL*. We further validate these experiments with a completely different model class, a neural network with linear layers and non-linear (ReLU) activations, varying the $\alpha$ weight decay regularizer parameter. Results for this model class can be found in Figures 9 and 13 in the Appendix.

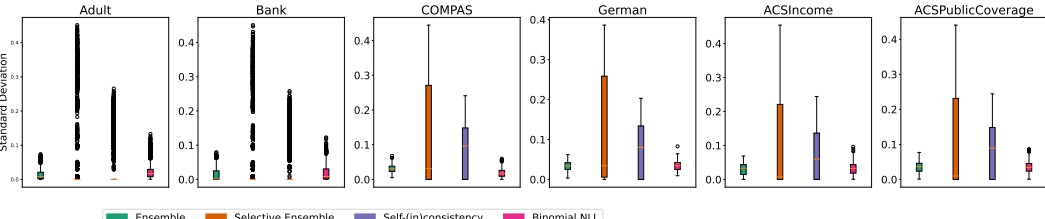

Figure 2: This boxplot shows the standard deviation of each individual's uncertainty estimates across different *max_depth* hyperparameter settings. For example, if an individual has the same uncertainty estimate for each hyper-parameter setting, then their standard deviation is 0 (perfect *consistency*) whereas if they vary wildly, the standard deviation is high (not *consistent*). The *Binomial NLL* and *Ensemble* methods exhibits are the most *consistent*.

Table 1: Comparison of *calibration* and *consistency* for each algorithm on each binary dataset. Here, calibration is measured by negative log-likelihood while the consistency is measured by the maximum individual empirical standard deviation. The $\pm$ indicates the standard deviation of these values over 10 iterations. Note that we adopt the Olympic medal convention in all Tables throughout our paper: gold , silver and bronze cells signify first, second and third best performance, respectively.

| | Calibration (Negative Log-Likelihood) | | | | | Consistency | | | | |
| Approach | ACS | Adult | Bank | COMPAS | German | ACS | Adult | Bank | COMPAS | German |
|---|---|---|---|---|---|---|---|---|---|---|
| *Ensemble* | $1.1 \pm 0.04$ | $0.9 \pm 0.02$ | $0.5 \pm 0.02$ | $1.8 \pm 0.07$ | $0.97 \pm 0.12$ | $0.08 \pm 0.00$ | $0.08 \pm 0.01$ | $0.09 \pm 0.01$ | $0.06 \pm 0.00$ | $0.062 \pm 0.00$ |
| *Selective Ens.* | $1.1 \pm 0.05$ | $0.88 \pm 0.02$ | $0.5 \pm 0.02$ | $1.8 \pm 0.08$ | $1.0 \pm 0.16$ | $0.45 \pm 0.01$ | $0.45 \pm 0.01$ | $0.45 \pm 0.01$ | $0.44 \pm 0.01$ | $0.40 \pm 0.02$ |
| *(In)cons. Ens.* | $1.0 \pm 0.04$ | $0.82 \pm 0.03$ | $0.42 \pm 0.02$ | $1.5 \pm 0.07$ | $0.82 \pm 0.13$ | $0.26 \pm 0.01$ | $0.26 \pm 0.01$ | $0.25 \pm 0.01$ | $0.25 \pm 0.01$ | $0.21 \pm 0.01$ |
| *Binom. NLL* | $0.4 \pm 0.01$ | $0.31 \pm 0.0$ | $0.2 \pm 0.0$ | $0.6 \pm 0.01$ | $0.5 \pm 0.04$ | $0.10 \pm 0.01$ | $0.13 \pm 0.01$ | $0.12 \pm 0.01$ | $0.07 \pm 0.00$ | $0.08 \pm 0.01$ |

**Calibration (qualitative)** The next experiment qualitatively evaluates calibration. For each method, we identify groups of individuals with similar uncertainty estimates and empirically evaluate the standard deviation of the residual difference between the observed and predicted outcomes. Results in Figure 3 demonstrate that predicted uncertainty estimates from the *Selective Ensemble* and *Self-(in)consistency* algorithms do not appear related to the empirical standard deviation. In contrast, the estimates from the *Ensemble* and *Binomial NLL* methods are clearly related to the empirical standard deviation; we note that the *Binomial NLL* method most closely tracks the identity line.

**Calibration (quantitative)** Finally, we quantitatively evaluate calibration by interpreting uncertainty estimates as the parameter of a fixed distribution. In the binary setting, the distribution is completely described by a single probability and the Binomial distribution. Table 1 gives the negative log-likelihood for each method on each dataset. To produce Table 1, we interpret estimates from models that produce a different kind of uncertainty as a standard deviation for comparison. By design, the *Binomial NLL* method minimizes the negative log-likelihood on the training set, so unsurprisingly we find that the *Binomial NLL* method also gives the best performance on the test set. Overall, Figures 2 and 3 and Table 1 suggest that the *Binomial NLL* method produces heteroscedastic uncertainty estimates that are simultaneously the most *consistent* and *calibrated*.

## 4.2 REGRESSION TASKS

In this section, we turn our attention to evaluating methods for estimating heteroscedastic uncertainty in the regression setting over continuous outcomes. `FairlyUncertain` contains several algorithms drawn from the heteroscedastic uncertainty literature.

- The *Normal NLL* method learns parameters $\mu$ and $\sigma$ that minimize the negative log-likelihood loss of the normal distribution, given by $\ell(y, \mu, \sigma) = -\log \sigma + \frac{1}{2} \left( \frac{x-\mu}{\sigma} \right)^2$.

- The $\beta$-*NLL* method (Seitzer et al., 2022) learns a version of the NLL loss that is multiplied by the constant $\sigma^{2\beta}$ (in our experiments we set $\beta = \frac{1}{2}$ as suggested in the original paper).

- The *Faithful NLL* method (Stirn et al., 2023) learns mean predictions $\mu$ with the standard mean squared error loss while the standard deviation predictions $\sigma$ are learned with the NLL loss.

- A natural *Ensemble* method serves as a point of comparison for these heteroscedastic algorithms: *Ensemble* learns an ensemble of models trained on bagged samples of the training set, and outputs the mean and standard deviation over model predictions.

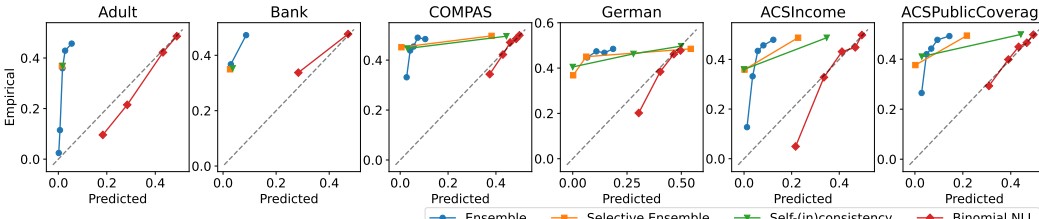

Figure 3: For five groups assembled by predicted uncertainty, we plot the average predicted uncertainty against the empirical standard deviation of the outcomes. An algorithm is perfectly calibrated if predicted uncertainty equals the empirical standard deviation i.e., the points lie on the dashed identity line. Note that uncertainty estimates do not always represent variance, so we expect a positive but not necessarily linear correlation. Additionally, note that this *calibration* graph also reflects *consistency*; a less *consistent* method will have a more arbitrary grouping leading to a flatter observed slope.

Table 2: Comparison of Negative Log-Likelihood (NLL) and Consistency for each algorithm on each dataset. We measure *consistency* as the maximum over individuals of the standard deviation of the predictions they receive with respect to the depth of the model.

| | Calibration (Negative Log-Likelihood) | | | | | | Consistency | | | | |
|---|---|---|---|---|---|---|---|---|---|---|---|
| Approach | Comm. | IHDP | Insur. | Law | Twins | | Comm. | IHDP | Insur. | Law | Twins |
| *Ensemble* | $120.0 \pm 21.0$ | $3.9 \pm 1.3$ | $21.0 \pm 15.0$ | $200.0 \pm 21.0$ | $160.0 \pm 9.3$ | | $0.05 \pm 0.01$ | $0.02 \pm 0.00$ | $0.03 \pm 0.00$ | $0.05 \pm 0.01$ | $0.09 \pm 0.01$ |
| *Normal NLL* | $0.35 \pm 0.08$ | $-0.6 \pm 0.07$ | $-0.67 \pm 0.04$ | $-0.25 \pm 0.0$ | $0.41 \pm 0.0$ | | $0.12 \pm 0.01$ | $0.08 \pm 0.01$ | $0.09 \pm 0.01$ | $0.11 \pm 0.01$ | $0.20 \pm 0.01$ |
| *$\beta$-NLL* | $0.34 \pm 0.08$ | $-0.6 \pm 0.02$ | $-0.65 \pm 0.03$ | $-0.25 \pm 0.0$ | $0.41 \pm 0.0$ | | $0.11 \pm 0.01$ | $0.06 \pm 0.01$ | $0.06 \pm 0.01$ | $0.08 \pm 0.00$ | $3.26 \pm 7.66$ |
| *Faithful NLL* | $0.34 \pm 0.11$ | $-0.58 \pm 0.01$ | $-0.64 \pm 0.03$ | $-0.25 \pm 0.0$ | $0.41 \pm 0.0$ | | $0.12 \pm 0.01$ | $0.08 \pm 0.01$ | $0.07 \pm 0.01$ | $0.09 \pm 0.01$ | $0.19 \pm 0.01$ |

Using `FairlyUncertain`, we evaluate these methods for *consistency* and *calibration*. We evaluate *consistency* in the same manner in the regression setting as we did in the binary setting, varying two hyperparameters while checking that the maximum deviation of uncertainty estimates remains stable (see Figure 10 in the Appendix). In Table 2, we see that the *Ensemble* approach is the most *consistent*, although for four of the five datasets, *$\beta$-NLL* performs similarly.

*Calibration* is also evaluated in the same manner for regression tasks as for binary. Figure 11 in the Appendix provides the qualitative assessment, while quantitatively Table 2 gives the NLL for each algorithm on each dataset. Our results demonstrate that though the *Ensemble* method is *consistent*, its predictions are very poorly *calibrated*. In contrast, all NLL approaches achieve a similar level of *calibration* across all five regression datasets. Overall, the various NLL approaches are all comparably *consistent* to the *Ensemble* method and are significantly more *calibrated*.

## 5 ABSTAINING ON CLASSIFICATION TASKS

We now turn our attention to incorporating uncertainty estimates into *fair* algorithms for classification, to assess their impact on downstream fairness metrics, focusing on the abstention framework where models are allowed to *abstain* from making a prediction if the heteroscedastic uncertainty estimates are too large (Black et al., 2022a; Cooper et al., 2024). `FairlyUncertain` instantiates this intervention, allowing models to abstain from predicting under high uncertainty. We find that while the abstention framework allows models to abstain from incorrect predictions, unsurprisingly reducing the overall error rate, it does not make the distribution of predictions more balanced between demographic groups. Moreover, the flexibility afforded by abstention is also its biggest limitation; if allowed to make almost no predictions, a model can easily achieve optimal performance without meaningful outputs for the majority of observations. Thus, `FairlyUncertain` crucially focuses on the question of abstention *rate* and how it affects overall error, statistical parity, equalized odds, etc.

In Figure 5, we see that allowing models to make fewer predictions based on uncertainty estimates decreases their error. In contrast, a *Random* baseline (real binary predictions but *random* uncertainty estimates) maintains the same error rate. Notice that the *Selective Ensemble* and *Self-(in)consistency Ensemble* produce the same uncertainty estimates for many observations, hence the flat lines. While abstaining invariably reduces overall error, it has a more chaotic impact on *Statistical Parity*. Figure 4 shows that abstaining can improve (decrease) or worsen (increase) statistical parity unpredictably across different data distributions; here, the model behavior resembles the *Random* uncertainty prediction baseline. We observe a similar trend in Figure 15, which plots the *Equalized Odds* fairness metric (Hardt et al., 2016) against the abstention rate (see Appendix E).

Our benchmark additionally evaluates uncertainty methods against standard baseline and state-of-the-art fairness algorithms. To construct this comparison, `FairlyUncertain` tests the *Random*,

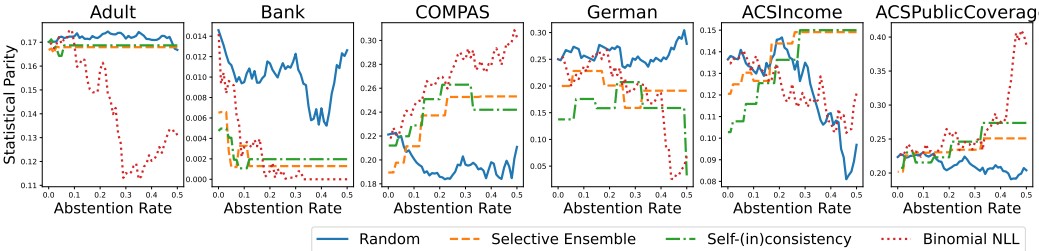

Figure 4: Abstaining has no reliable effect on *Statistical Parity* (comparable to the *Random* baseline).

Table 3: Various evaluation and fairness metrics for each algorithm on the ACS Income (see Appendix I for definitions). While the abstention framework allows models to reduce their *Error Rate*, it does *not* magically reduce the imbalance in outcomes between demographic groups.

| Approach | Error Rate | Statistical Parity | Equalized Odds | Equal Opportunity | Disparate Impact | Predictive Parity | False Positive Rate | Included % |
|---|---|---|---|---|---|---|---|---|
| Baseline | 0.22 ± 0.009 | 0.181 ± 0.023 | 0.165 ± 0.049 | 0.165 ± 0.049 | 1.65 ± 0.136 | 0.057 ± 0.034 | 0.083 ± 0.024 | 100.0 ± 0.0 |
| Threshold Opt. SP | 0.233 ± 0.009 | 0.029 ± 0.021 | 0.135 ± 0.046 | 0.063 ± 0.033 | 0.979 ± 0.07 | 0.241 ± 0.043 | 0.134 ± 0.046 | 100.0 ± 0.0 |
| Threshold Opt. EO | 0.226 ± 0.01 | 0.101 ± 0.03 | 0.08 ± 0.049 | 0.079 ± 0.049 | 1.3 ± 0.113 | 0.119 ± 0.039 | 0.024 ± 0.021 | 100.0 ± 0.0 |
| Exponentiated Grad. SP | 0.229 ± 0.011 | 0.041 ± 0.022 | 0.117 ± 0.038 | 0.065 ± 0.04 | 0.946 ± 0.08 | 0.203 ± 0.033 | 0.115 ± 0.038 | 100.0 ± 0.0 |
| Exponentiated Grad. EO | 0.218 ± 0.008 | 0.101 ± 0.041 | 0.092 ± 0.049 | 0.09 ± 0.052 | 1.3 ± 0.15 | 0.119 ± 0.045 | 0.031 ± 0.017 | 100.0 ± 0.0 |
| Grid Search SP | 0.23 ± 0.012 | 0.083 ± 0.045 | 0.142 ± 0.064 | 0.108 ± 0.067 | 1.04 ± 0.259 | 0.183 ± 0.048 | 0.106 ± 0.067 | 100.0 ± 0.0 |
| Grid Search EO | 0.227 ± 0.008 | 0.148 ± 0.034 | 0.132 ± 0.07 | 0.125 ± 0.074 | 1.48 ± 0.152 | 0.087 ± 0.048 | 0.056 ± 0.031 | 100.0 ± 0.0 |
| FairGBM SP | 0.216 ± 0.007 | 0.046 ± 0.041 | 0.049 ± 0.027 | 0.029 ± 0.026 | 1.13 ± 0.128 | 0.118 ± 0.062 | 0.038 ± 0.025 | 100.0 ± 0.0 |
| FairGBM EO | 0.215 ± 0.008 | 0.142 ± 0.027 | 0.124 ± 0.035 | 0.112 ± 0.046 | 1.47 ± 0.123 | 0.076 ± 0.045 | 0.071 ± 0.033 | 100.0 ± 0.0 |
| Random SP | 0.215 ± 0.011 | 0.142 ± 0.024 | 0.14 ± 0.043 | 0.138 ± 0.046 | 1.49 ± 0.109 | 0.07 ± 0.05 | 0.062 ± 0.031 | 89.1 ± 7.08 |
| Ensemble SP | 0.18 ± 0.017 | 0.159 ± 0.036 | 0.122 ± 0.055 | 0.122 ± 0.055 | 1.58 ± 0.19 | 0.052 ± 0.039 | 0.064 ± 0.025 | 81.7 ± 7.58 |
| Selective Ensemble SP | 0.17 ± 0.013 | 0.17 ± 0.035 | 0.152 ± 0.059 | 0.152 ± 0.06 | 1.69 ± 0.21 | 0.069 ± 0.045 | 0.06 ± 0.028 | 83.2 ± 5.06 |
| Self-(in)consistency SP | 0.176 ± 0.028 | 0.162 ± 0.039 | 0.156 ± 0.051 | 0.156 ± 0.051 | 1.63 ± 0.223 | 0.078 ± 0.047 | 0.049 ± 0.026 | 86.6 ± 7.95 |
| Binomial NLL SP | 0.156 ± 0.013 | 0.164 ± 0.035 | 0.158 ± 0.063 | 0.157 ± 0.066 | 1.67 ± 0.216 | 0.076 ± 0.061 | 0.044 ± 0.029 | 77.3 ± 3.07 |
| Random EO | 0.215 ± 0.01 | 0.147 ± 0.024 | 0.128 ± 0.045 | 0.126 ± 0.047 | 1.51 ± 0.109 | 0.064 ± 0.047 | 0.067 ± 0.028 | 90.2 ± 8.45 |
| Ensemble EO | 0.174 ± 0.011 | 0.17 ± 0.03 | 0.108 ± 0.052 | 0.107 ± 0.053 | 1.64 ± 0.17 | 0.059 ± 0.046 | 0.068 ± 0.028 | 77.8 ± 2.82 |
| Selective Ensemble EO | 0.17 ± 0.013 | 0.171 ± 0.035 | 0.147 ± 0.058 | 0.147 ± 0.058 | 1.69 ± 0.208 | 0.071 ± 0.044 | 0.061 ± 0.027 | 83.3 ± 5.0 |
| Self-(in)consistency EO | 0.174 ± 0.022 | 0.167 ± 0.04 | 0.154 ± 0.062 | 0.154 ± 0.062 | 1.67 ± 0.231 | 0.064 ± 0.045 | 0.056 ± 0.027 | 86.0 ± 6.29 |
| Binomial NLL EO | 0.171 ± 0.023 | 0.158 ± 0.03 | 0.135 ± 0.054 | 0.134 ± 0.055 | 1.61 ± 0.177 | 0.07 ± 0.054 | 0.051 ± 0.028 | 82.8 ± 7.44 |

*Selective Ensemble*, *Self-(in)consistency*, and *Binomial NLL* methods at inclusion rates between 75% and 100%. It selects an abstention rate for comparison by optimizing a simple objective function that is the normalized sum of the *Error Rate*, *Statistical Parity*, and *Equalized Odds*. Table 3 gives performances on the ACS Income task (Ding et al., 2021) (results for the other datasets appear in Appendix F). Once again, methods allowed to abstain had lower error (*Self-(in)consistency* and *Binomial NLL* methods had the lowest). However, across 5 of the 6 fairness metrics reported, methods that did not abstain were the *highest* performing in terms of fairness. This is surprising, as it seems directly opposed to one of the main claims in Cooper et al. (2024).

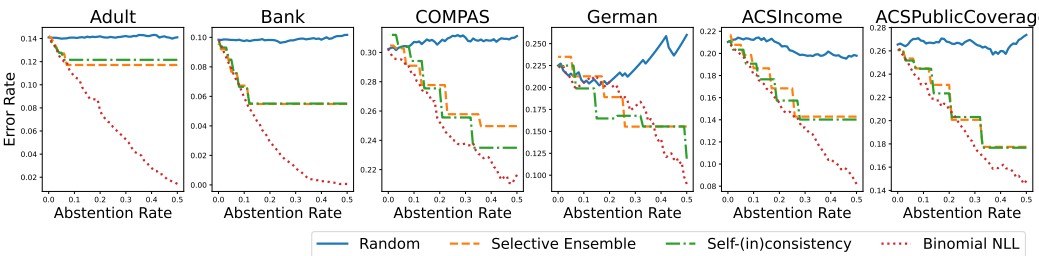

Figure 5: For an abstention rate $r$, FairlyUncertain abstains on the $r$ fraction of observations with the highest uncertainty. For heteroscedastic uncertainty methods, predictions become more accurate as the model abstains more, while the error rate for the random baseline remains steady.

A natural question arises given the results in Table 3: for which individuals do the abstaining models choose not to predict? In Figure 17 (deferred to Appendix E), we address this by comparing the empirical distributions of the overall population and the included (non-abstained) population for each feature across methods. We compute the average Wasserstein distance between these distributions (averaged over methods) and plot the feature with the largest distance. The variables with the greatest differences tend to be protected attributes like marriage status and sex. However, except for the *Binomial NLL* distribution on the *Adult* dataset, these differences are small between the overall distribution and the one selected through the abstaining process.

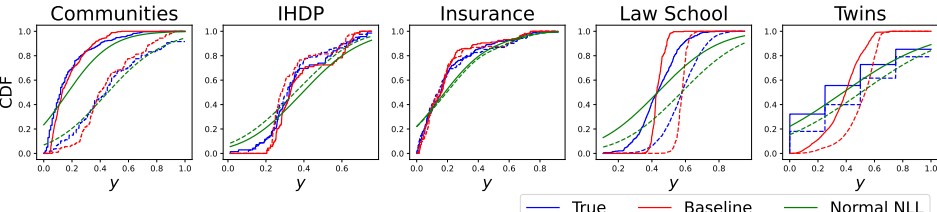

Figure 6: CDF by group of on each dataset. The solid lines indicate one protected group and the dashed lines indicate the other protected group. Normal NLL effectively smooths the CDF.

# 6 Uncertainty Aware Fair Regression

So far, we have used `FairlyUncertain` to demonstrate that incorporating heteroscedastic uncertainty estimates into *fair* classification algorithms requires care; this is no different in the regression setting. By outputting both predictions and uncertainty estimates in either setting, downstream users have the flexibility to apply uncertainty estimates in domain-appropriate ways *beyond abstention*. For example, practitioners can flag high uncertainty estimates for manual review (Madras et al., 2018), combine predictions and uncertainty estimates into domain-specific metrics (Ali et al., 2021), or discover patterns of high uncertainty that indicate a lack of meaningful features, prompting more data collection (Chen et al., 2018). Still, it remains an open question how to measure the fairness for models that output *both* predictions and uncertainty estimates, particularly for regression models. To address this, we propose a new fairness metric (included in `FairlyUncertain`), *uncertainty-aware statistical parity (UA-SP)* (Definition 6.2), which naturally incorporates the available uncertainty estimates into the the standard regression-specific form of *Statistical Parity* (Definition 6.1).

Note that *UA-SP* (Definition 6.2) applies to models which produce point predictions $\mu$ *and* uncertainty estimates $\sigma$ for each observation. These estimates are treated as model-derived inputs to our fairness metric, not as the underlying true parameters. We model predictions as distributions to incorporate the inherent uncertainty of model outputs; this approach is particularly salient in settings with heteroscedastic variance. When the uncertainty estimates are zero, our metric reduces to the standard statistical parity definition, thus generalizing it to account for variability in predictions due to uncertainty. The generalized definition assumes that the uncertainty follows a normal distribution with mean $\mu$ and standard deviation $\sigma$. The validity of this assumption depends on the setting.

**Definition 6.1** (Statistical Parity). *Consider a method $f : \mathcal{X} \times \mathcal{A} \to \mathcal{Y}$ that outputs a mean prediction. Then $f$ satisfies statistical parity if $\Pr(f(\mathbf{X}, A) \geq y | A = a) = \Pr(f(\mathbf{X}, A) \geq y)$ for all protected groups $a \in \mathcal{A}$ and outcomes $y \in \mathcal{Y}$.*

**Definition 6.2** (Uncertainty-Aware Statistical Parity (*UA-SP*)). *Consider $f : \mathcal{X} \times \mathcal{A} \to \mathcal{Y} \times \mathbb{R}$ that estimates a mean $\mu$ and a standard deviation $\sigma$. The predictions induce a randomized function $\tilde{f} : \mathcal{X} \times \mathcal{A} \to \mathcal{Y}$ that samples $y \sim \mathcal{N}(\mu, \sigma^2)$. Then $f$ satisfies uncertainty-aware statistical parity if $\Pr(\tilde{f}(\mathbf{X}, A) \geq y | A = a) = \Pr(\tilde{f}(\mathbf{X}, A) \geq y)$ for all protected groups $a \in \mathcal{A}$ and outcomes $y \in \mathcal{Y}$.*

Definition 6.2 incorporates uncertainty estimates in a natural way: the predicted distribution is smoothed for observations with high uncertainty. In contrast, Definition 6.1 holds the algorithms to a stringent standard even when heteroscedastic variance is large.

In regression settings, Definition 6.1 is often interpreted in terms of a Cumulative Density Function (CDF) over predictions (Agarwal et al., 2019; Liu et al., 2022); uncertainty-aware statistical parity (Definition 6.2) shares this interpretation. For methods that only produce predictions, the CDF is with respect to the randomness of the observations $\mathbf{x}$. For methods that produce predictions *and* uncertainty estimates, the CDF is with respect to the randomness of the observations and the induced randomized function $\tilde{f}$. Figure 6 gives CDFs for *True*, *Baseline* and *Normall NLL*: here, *True* indicates the correct labels in the test data, *Baseline* indicates a standard regression model without any fairness intervention, and *Normal NLL* is the smoothed method as described in Section 4.2 (for visual clarity, we present comparisons only against *Normal NLL*, which is similar to *Faithful NLL* and more *consistent* and *calibrated* than *Ensemble* and $\beta$-*NLL*).

Figure 6 illustrates how incorporating uncertainty into model predictions effectively smooths the CDF. In the case of the Twins dataset, we observe that this smoothing is both strong and accurate, allowing the model to closely capture the true distribution of the labels. Table 4 presents fairness metrics computed according to either Definition 6.1 or Definition 6.2, depending on whether the model provides uncertainty estimates. The *Normal NLL* method significantly outperforms others in terms of fairness, even surpassing the *Ensemble* method. Remarkably, the *Normal NLL* method, which is both competitively *consistent* and *calibrated*, achieves substantial fairness improvements *without* any explicit fairness interventions.

## 6.1 Related Work

Prior work in algorithmic fairness largely focuses on fixed fairness constraints, framing the issue as optimizing along a Pareto frontier between accuracy and fairness metrics that may conflict (Hardt

Table 4: Measuring (uncertainy-aware) statistical parity (*UA-SP*) via Kolmogorov-Smirnov distance: the maximum over outputs $y$ of the distance between per-group CDFs (Agarwal et al., 2019). Simply using *consistent* and *calibrated* uncertainty methods substantially reduces *UA-SP*.

| Approach | Communities | IHDP | Insurance | Law School | Twins |
|---|---|---|---|---|---|
| True | $0.561 \pm 0.036$ | $0.153 \pm 0.061$ | $0.1 \pm 0.022$ | $0.484 \pm 0.021$ | $0.161 \pm 0.011$ |
| Baseline | $0.644 \pm 0.025$ | $0.144 \pm 0.05$ | $0.091 \pm 0.019$ | $0.965 \pm 0.009$ | $0.404 \pm 0.009$ |
| Exponentiated Gradient Square | $0.638 \pm 0.024$ | $0.16 \pm 0.053$ | $0.094 \pm 0.025$ | $0.902 \pm 0.015$ | $0.357 \pm 0.01$ |
| Exponentiated Gradient Absolute | $0.638 \pm 0.024$ | $0.157 \pm 0.057$ | $0.094 \pm 0.025$ | $0.902 \pm 0.015$ | $0.359 \pm 0.009$ |
| Grid Search Square | $0.649 \pm 0.035$ | $0.167 \pm 0.056$ | $0.093 \pm 0.025$ | $0.888 \pm 0.019$ | $0.357 \pm 0.007$ |
| Grid Search Absolute | $0.649 \pm 0.035$ | $0.167 \pm 0.056$ | $0.092 \pm 0.024$ | $0.888 \pm 0.019$ | $0.358 \pm 0.007$ |
| Ensemble | $0.62 \pm 0.021$ | $0.101 \pm 0.052$ | $0.076 \pm 0.022$ | $0.939 \pm 0.008$ | $0.389 \pm 0.009$ |
| Normal NLL | $0.37 \pm 0.016$ | $0.048 \pm 0.028$ | $0.052 \pm 0.022$ | $0.196 \pm 0.003$ | $0.096 \pm 0.002$ |
| $\beta$-NLL | $0.375 \pm 0.019$ | $0.045 \pm 0.025$ | $0.051 \pm 0.022$ | $0.195 \pm 0.004$ | $0.096 \pm 0.003$ |
| Faithful NLL | $0.299 \pm 0.151$ | $0.042 \pm 0.021$ | $0.054 \pm 0.02$ | $0.192 \pm 0.004$ | $0.095 \pm 0.003$ |

et al., 2016; Zafar et al., 2017; Mitchell et al., 2021; Bell et al., 2023; Wang et al., 2024b). More recent research has introduced *uncertainty* estimation as a crucial factor in model selection under fairness. Particularly relevant are Black et al. (2022a) and Cooper et al. (2024), who explore ensemble approaches to estimate prediction uncertainty based on the standard deviation from an ensemble of models. These methods, as we have demonstrated, tend to focus on artifacts of the learning pipeline rather than inherent heteroscedastic uncertainty.

Other studies examine uncertainty estimation through model multiplicity (Black et al., 2022b), analyze the relationship between inadequate sample sizes and disparate epistemic uncertainty among subgroups (Chen et al., 2018; Zhang & Long, 2021), and extend abstention frameworks to regression (Shah et al., 2022). Additionally, tailored inference and prediction models under heteroscedastic assumptions have been extensively studied in economics and statistics, particularly with Bayesian inference (White, 1980; MacKinnon, 2012; Rigobon, 2003; Hayes & Cai, 2007; Ji et al., 2020). The impact of Bayesian conditioning via latent variables on fairness has also been considered as an alternative to bagging for uncertainty estimation, though *calibration* of these estimates can be challenging due to the large space of priors (McNair, 2018; Ji et al., 2020). More recently, loss functions accommodating heteroscedastic assumptions in machine learning have improved uncertainty estimation and model robustness (Collier et al., 2020; Abdar et al., 2021), with some work addressing heteroscedastic pitfalls in log-likelihood-based loss functions (Seitzer et al., 2022; Stirn et al., 2023).

Prior work on uncertainty in fair regression has focused on estimating quantiles – confidence intervals – for predictions (Liu et al., 2022; Kuzucu et al., 2023; Wang & Wang, 2024). Quantile predictions are considered fair if they are equally calibrated and accurate for different demographic groups (Wang et al., 2024b;a). However, quantiles are only valid for a given probability threshold and fail to adequately describe the distribution. For example, recall how Figure 1 demonstrated how the mean and quantiles could be the same for two distributions that are substantially different.

## 7 CONCLUSION

`FairlyUncertain` actualizes the *consistency* and *calibration* axioms in the form of a robust benchmark for evaluating uncertainty estimates in fair predictions. `FairlyUncertain` suggests the following results: In the binary setting, natural uncertainty estimates beat complex ensemble based approaches and abstaining improves error but not imbalance between demographic groups. In the regression setting, *consistent* and *calibrated* uncertainty methods can reduce distributional imbalance without any explicit fairness intervention. The current version of `FairlyUncertain` is not without limitations: different models (Gorishniy et al., 2021), parameters, and datasets could be varied to assess *consistency*, different metrics introduced for *calibration*, and additional fairness interventions explored. Because of its specialized nature, we expect a net positive social impact from `FairlyUncertain`. We hope that the extensible package construction allows `FairlyUncertain` to grow with the community.

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

# A   ADDITIONAL COMMENTS ON UNCERTAINTY AND FAIRNESS

As methods for uncertainty estimation in prior work differ on the actual estimation objective (to varying degrees), we find it useful to provide a precise typology over sources of uncertainty (Hüllermeier & Waegeman, 2021) in Table 5. This typology helps us to define precise targets for each uncertainty estimation evaluation strategy in our benchmark.

As an example of how this typology helps describe reasonable uncertainty estimate objectives, consider the scenario given in Figure 1, which exemplifies both **(A)** and **(B)** uncertainty in Table 5. Producing a *fair* uncertainty estimate ultimately lies in understanding **(A)** and **(B)** uncertainties (Black et al., 2022a; Cooper et al., 2024), but empirical methods over a fixed data sample can only directly estimate **(C)**, **(D)**, and **(E)**. A goal of our work is to use estimates of **(C)**-**(E)** to assess the *consistency* and *calibration* of approximations for the unmeasurable uncertainties **(A)** and **(B)**.

Table 5: The following *uncertainty typology* allows us to speak precisely about sources of uncertainty we can estimate directly from data, and sources of uncertainty we can only hope to approximate.

| Uncertainty Type | Description | Example |
|---|---|---|
| **(A) Unmeasurable individual-level uncertainty** | Uncertainty inherent to individual outcomes that cannot be measured. | A student may experience a unique, random moment of distraction during a test. |
| **(B) Within-individual variability** | Uncertainty captured through repeated measurements of an individual. | Scores for one student on tests throughout the year across different testing conditions. |
| **(C) Across-individual variability** | Uncertainty arising from differences between individuals (covariate attributable). | Two students perform differently due to factors such as access to study resources, etc. |
| **(D) Sampling uncertainty** | Uncertainty stemming from repeated sampling i.e. process of data collection. | Differing test scores between two random samples of students from sample variability. |
| **(E) Modeling uncertainty** | Uncertainty introduced by the modeling process itself (hyperparameters, etc.). | Different models give slightly different predictions for the same set of students. |

**Societal Implications**   Predictive uncertainty estimation is not merely a technical consideration but has profound implications for fairness and justice in algorithmic decision-making (Bhatt et al., 2021; Cooper et al., 2024). Our research contributes the following insight: naively incorporating uncertainty into fair models can lead to unpredictable and potentially adverse outcomes for certain demographic groups. For example, our empirical analyses demonstrate that while abstention methods - where models defer decisions under high uncertainty - can reduce overall error rates, they do not necessarily improve fairness metrics such as statistical parity or equalized odds. This unpredictability may exacerbate existing disparities and undermine trust in these systems. Furthermore, the arbitrary application of uncertainty estimates might violate anti-discrimination laws and regulations. Uncertainty estimation is also integral to procedural justice (Rawls, 2017), which concerns the fairness of the methods and procedures used to arrive at decisions. By advocating for uncertainty estimates that are consistent across similar models and calibrated to actual data variability, we provide a more robust foundation for ethical algorithmic decision-making.

# B  FAIRLYUNCERTAIN: DATASETS AND EXPERIMENTAL DETAILS

**Experimental Details**   We use a cluster of 24-core Intel Cascade Lake Platinum 8268 chips to run the experiments. We use default model hyperparameters except when the experiment explicitly varies them i.e., for assessing *consistency*. Our benchmark considers a privileged class and an unprivileged class for each dataset. The privileged class can be at the intersection of multiple protected features; for example, "white" (race) and "male" (gender) could become a privileged class subset of "white males" for evaluation. All fairness metrics in the submitted paper are then binary and are the absolute difference between the disadvantaged and advantaged group. The benchmark defaults to using the XGBoost model for all predictive tasks; XGBoost is a fast, state-of-the-art model that generally outperforms neural models in relatively low data, low dimensional tabular regimes, like most datasets in our benchmark (Chen & Guestrin, 2016; Grinsztajn et al., 2022; McElfresh et al., 2024).

Table 6: We note that even though ACS Folktables is the improved version of Adult, we include Adult for parity with prior work (Ding et al., 2021; Cooper et al., 2024); we include COMPAS and Communities & Crimes for similar parity reasons, although highlight significant concerns with automating criminal justice (Fabris et al., 2022; Thomas & Pontón-Núñez, 2022).

| Datasets (binary) | Size | # Feat. | Goal is to predict... | Protected Att. |
|---|---|---|---|---|
| ACS Ding et al. (2021) | 16249 | 16 | whether an individual is employed | race |
| Adult Kohavi et al. (1996) | 45222 | 102 | whether individual income exceeds a certain level | gender |
| Bank Marketing Moro et al. (2014) | 30488 | 57 | whether clients will subscribe to a product | age |
| COMPAS Angwin et al. (2022) | 6167 | 406 | whether a defendant will re-offend | gender |
| German Credit Hofmann (2000) | 1000 | 57 | whether an individual has 'good' or 'bad' credit | age |
| **Datasets (regression)** | | | | |
| Law School GPA Sander (2004) | 22342 | 4 | students' GPA in law school | race |
| Communities & Crimes Redmond & Baveja (2002) | 1994 | 100 | # of per-capita violent crimes in a community | race |
| Insurance Lantz (2019) | 1338 | 8 | individual medical costs billed by insurance | gender |
| IHDP Hill (2011) | 747 | 26 | the cognitive test scores of infants | gender |
| Twins Almond et al. (2005) | 68995 | 19 | the number of prenatal visits | race |

## C   EXAMPLE USAGE FAIRLYUNCERTAIN

```python
1  import fairlyuncertain as fu
2
3  algorithms = {algo_name: fu.algorithms[algo_name] for algo_name in fu.binary_uncertainty}
4
5  results = fu.get_calibration_table_data(is_binary=True, algorithms=algorithms, datasets=fu.binary_datasets, num_runs=10)
6
7  fu.print_table(results)
```

Figure 7: Complete code for producing the binary calibration table. The package is extensible, with easy to plug in datasets and algorithms. Each dataset simply needs to implement a loading function and each algorithm simply needs to output a prediction.

# D    ADDITIONAL RESULTS ON CONSISTENCY AND CALIBRATION

**Why to Use *Binomial NLL* for Binary Classification Uncertainty**    Consider a model $\tilde{f} : \mathcal{X} \times \mathcal{A} \to \mathcal{Y} \times \mathbb{R}$ that outputs binary predictions $\tilde{y}$ and uncertainty estimates $\tilde{\sigma}$. We can evaluate the NLL of the estimates by converting them to probabilities. We can solve for probabilities that produce the estimates $\tilde{\sigma}$ with the quadratic formula:[2]

$$\tilde{p}^- = \frac{1 - \sqrt{1 - 4\tilde{\sigma}^2}}{2} \qquad\qquad \tilde{p}^+ = \frac{1 + \sqrt{1 - 4\tilde{\sigma}^2}}{2}.$$

Implicitly, *Binomial NLL* argues that if we had some $(\tilde{\sigma}')^2$ that was more accurate than the estimated uncertainty $\tilde{\sigma}$, then we could get *more accurate probability predictions $\tilde{p}'$ from $(\tilde{\sigma}')^2$*. Thus, we should simply do our best to estimate $\tilde{p}$ (with NLL as the natural objective), which then induces a standard deviation $\tilde{\sigma}$. This motivates *Binomial NLL* and likely explains its strong performance.

**Additional experiments**    In Figure 8, we present boxplots of individual uncertainty predictions for binary classification, varying the reduction threshold $\gamma$ for the XGBoost model; the *Binomial NLL* method demonstrates greater consistency across different $\gamma$ values. Figure 10 shows similar boxplots in the regression setting, varying *max_depth* (top plot) and reduction threshold $\gamma$ (bottom plot), indicating that the *Ensemble* method is the most consistent across *max_depth* values, while other methods exhibit similar consistency levels. Figure 11 illustrates the calibration of various algorithms in regression by plotting the empirical standard deviation against the predicted uncertainty using 100 bins. Finally, Figure 12 displays uncertainty plots for binary classification across datasets and methods, varying *max_depth*, demonstrating that the *Ensemble* method produces consistent uncertainty estimates, whereas the *Selective Ensemble* method shows large variance and inconsistency in uncertainty estimates.

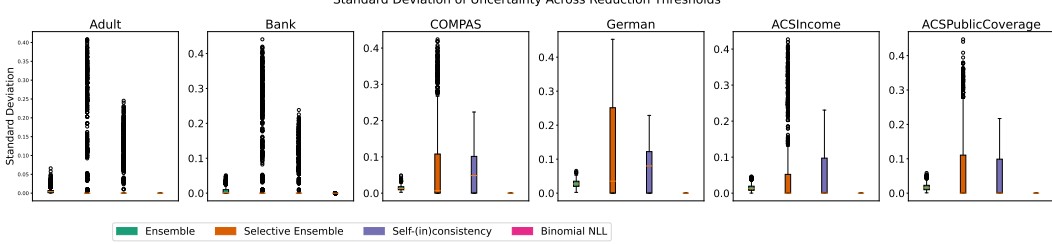

Figure 8: Boxplot displaying the mean, variance, and outliers of individual uncertainty predictions for the binary classification setting, with varying the *reduction_threshold* hyperparameter $\gamma$ (bottom plot) for the XGBoost model. The Binomial NLL method demonstrates significantly greater consistency across varying $\gamma$ values.

Table 7: Comparison of *calibration* for each algorithm on each binary dataset in terms of Expected Calibration Error (ECE) Naeini et al. (2015). Note that lower values are better.

| Approach | ACS | Adult | Bank | COMPAS | German |
|---|---|---|---|---|---|
| Ensemble | $1.08 \pm 0.037$ | $0.899 \pm 0.023$ | $0.499 \pm 0.018$ | $1.79 \pm 0.068$ | $0.968 \pm 0.12$ |
| Selective Ensemble | $1.09 \pm 0.048$ | $0.876 \pm 0.024$ | $0.501 \pm 0.024$ | $1.77 \pm 0.082$ | $1.05 \pm 0.156$ |
| Self-(in)consistency | $1.02 \pm 0.041$ | $0.819 \pm 0.027$ | $0.419 \pm 0.018$ | $1.52 \pm 0.068$ | $0.823 \pm 0.125$ |
| Binomial NLL | $0.396 \pm 0.008$ | $0.307 \pm 0.004$ | $0.198 \pm 0.004$ | $0.598 \pm 0.009$ | $0.503 \pm 0.036$ |

---

[2]The catch is that two probabilities can map to the same standard deviation. Since the model $\tilde{f}$ also makes predictions $\tilde{y}$, we will choose the probability $\tilde{p}$ closer to the prediction.

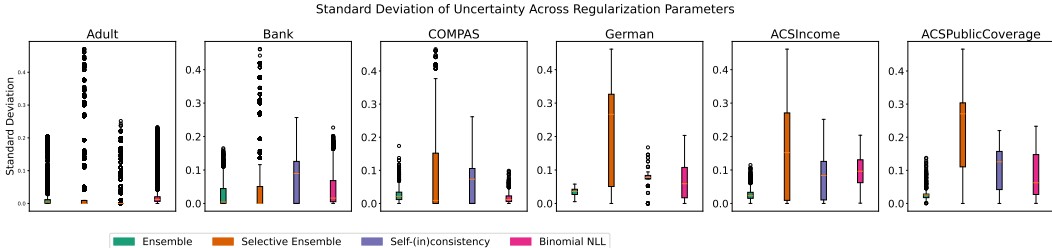

Figure 9: Boxplot displaying the mean, variance, and outliers of individual uncertainty predictions for the binary classification setting, with varying the $\alpha$ weight-decay regularization hyperparameter for the a neural net with linear layers and non-linear activations (e.g. ReLU). The Ensemble method demonstrates generally greater consistency across varying $\alpha$ values, followed by the Binomiall NLL method.

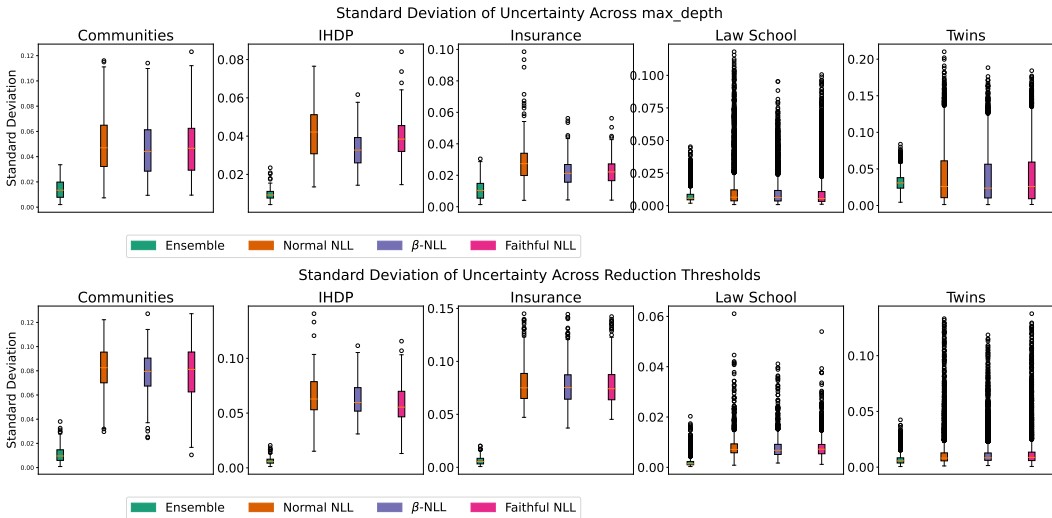

Figure 10: Boxplots showing the variance of individual uncertainty predictions for the **regression** setting with varying max_depth (top plot) and reduction threshold $\gamma$ (bottom plot) parameters. The Ensemble method is the most consistent across the max_depth values, while all other methods exhibit similar levels of consistency.

## D.1 ON CALIBRATION METRICS

Expected Calibration Error (ECE) Naeini et al. (2015) and Negative Log Likelihood (NLL) will differ fundamentally in their model calibration assessment. Given predictions $\{p_i\}_{i=1}^N$, uncertainty estimates

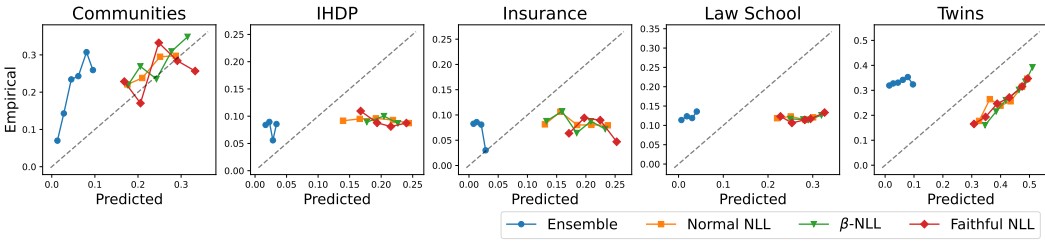

Figure 11: The *calibration* of various algorithms, using the XGBoost model. We compute *calibration* by making 100 bins on the uncertainty measure: we plot the empirical standard standard deviation in the bin against the predicted uncertainty.

$\{\sigma_i\}_{i=1}^N$, and true labels $\{y_i\}_{i=1}^N$, ECE groups predictions into $M$ confidence bins $\{B_m\}_{m=1}^M$ based on $p_i$, and computes calibration as $\text{ECE} = \sum_{m=1}^M \frac{|B_m|}{N} |\text{acc}(B_m) - \text{conf}(B_m)|$, where $\text{acc}(B_m) = \frac{1}{|B_m|} \sum_{i \in B_m} \mathbf{1}\{\hat{y}_i = y_i\}$ and $\text{conf}(B_m) = \frac{1}{|B_m|} \sum_{i \in B_m} p_i$.

In contrast, our modified NLL incorporates uncertainty estimates directly by adjusting predicted probabilities as $\tilde{p}_i = (p_i > 0.5) \cdot p_a + (p_i \leq 0.5) \cdot p_b$, where $p_a = (1 + \sqrt{1 - 4\sigma_i^2})/2$ and $p_b = (1 - \sqrt{1 - 4\sigma_i^2})/2$. We then compute NLL in standard fashion e.g. $\text{NLL} = -\frac{1}{N} \sum_{i=1}^N [y_i \log(\tilde{p}_i) + (1 - y_i) \log(1 - \tilde{p}_i)]$.

See Table 7 for results on the metric across each of our methods of calculating uncertainty. ECE provides an interpretable, aggregate view of calibration by measuring the alignment of predicted probabilities with empirical accuracy in confidence intervals (Naeini et al, 2015). However, it is sensitive to binning choices and lacks granularity at the individual prediction level. Our adjusted NLL method avoids binning, directly incorporating uncertainty estimates to evaluate calibration at a finer resolution, penalizing overconfident errors and underconfident correct predictions. While this makes NLL more sensitive to uncertainty quality, it may conflate calibration with model discrimination, and its dependence on predicted standard deviations assumes valid uncertainty estimates. We'd expect to prefer something like ECE for global calibration trends, while our NLL-based approach is suited to uncertainty-aware evaluation at the individual level.

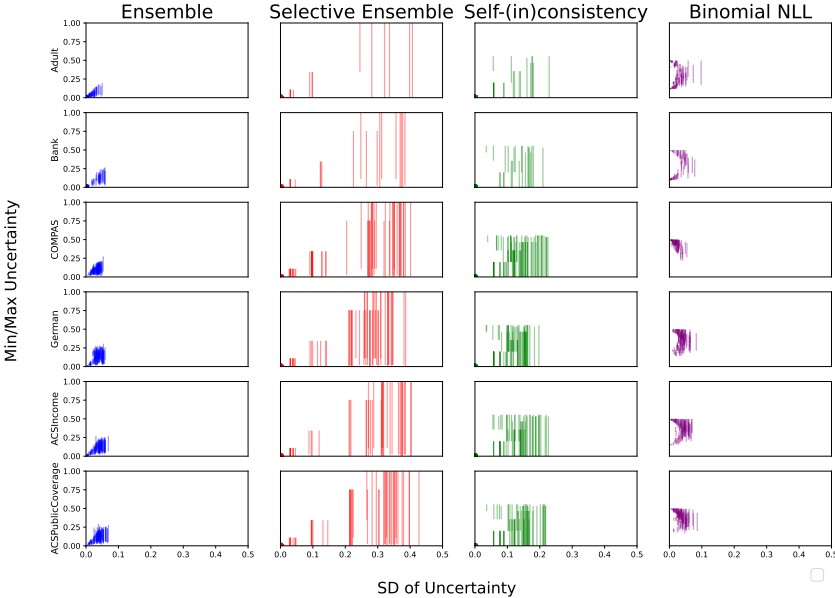

Figure 12: Plots of uncertainty for the binary classification problem across datasets (rows) and methods (columns), where we vary the hyperparameter *max-depth*. Each individual is plotted as a line, where the line's higher y-coordinate is the *maximum* uncertainty estimate over all hyperparameter settings for that, and the lower y-coordinate is the individual's *minimum* uncertainty estimate. Each individual is also placed along the $x$-axis at the standard deviation of their uncertainty estimates over the hyper-parameter settings. The *Ensemble* method produces *consistent* uncertainty estimates: the standard deviation of these estimates is less than 0.1 for all datasets and the uncertainty range starts close to 0 and increases with the standard deviation of the uncertainty (this is desirable). In contrast, the *Selective Ensemble* method produces *inconsistent* uncertainty estimates: the standard deviation of the uncertainty predictions is large and the range of the uncertainty predictions is large (especially for large standard deviations).

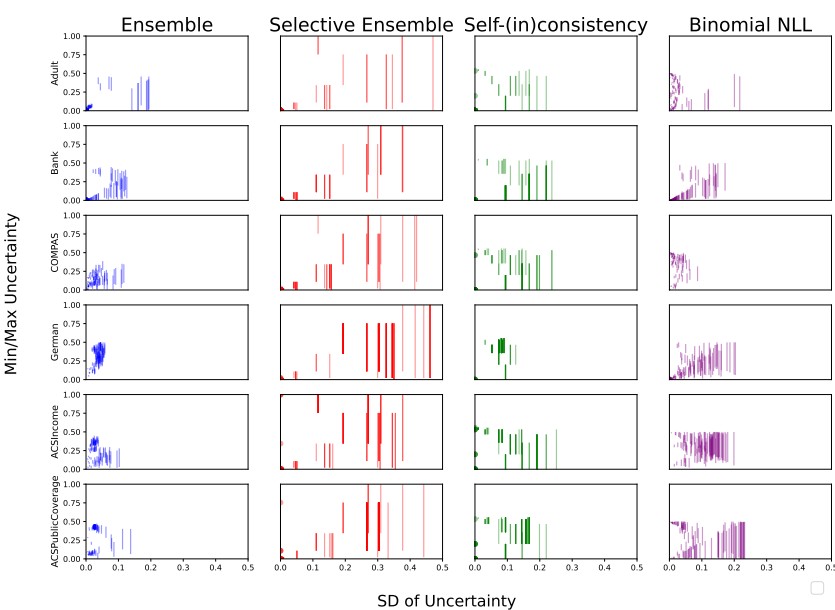

Figure 13: Similar plots of uncertainty for the binary classification problem across datasets (rows) and methods (columns) to Figure 12, but where the model is a neural network with linear layers and non-linear activations, and the parameter varying is $\alpha$ weight-decay regularization. Again, the *Ensemble* method produces *consistent* uncertainty estimates: the standard deviation of these estimates is less than 0.1 for all datasets and the uncertainty range starts close to 0 and increases with the standard deviation of the uncertainty (this is desirable).

# E BINARY ABSTENTION RESULTS

In Example E.1 below, we construct a predictive setting where abstaining can arbitrarily harm a standard fairness metric, like equalized odds.

**Example E.1** (Abstention Can Harm Fairness Metrics). *Consider a binary classification model evaluated on two demographic groups, A and B, each with $N_A = N_B$ examples. Both groups contain an equal number of positive ($Y = 1$) and negative ($Y = 0$) examples ($N_{A,1} = N_{A,0} = N_{B,1} = N_{B,0}$). Without abstention, the model predicts perfectly, achieving a true positive rate (TPR) and false positive rate (FPR) of $TPR_A = TPR_B = 1.0$ and $FPR_A = FPR_B = 0.0$, satisfying equalized odds. Say the model incorporates abstention based on uncertainty, and for group A, the model has arbitrarily low uncertainty and continues predicting perfectly ($TPR_A = 1.0, FPR_A = 0.0$). However, the model has high uncertainty for group B, and thus the model abstains on all positive examples ($Y = 1$), resulting in $TPR_B = 0.0$, while still predicting negatives correctly ($FPR_B = 0.0$). This abstention-induced disparity in TPRs ($TPR_A - TPR_B = 1.0$) violates the equalized odds fairness metric to an arbitrary degree.*

`FairlyUncertain` explores how abstaining from binary predictions based on uncertainty affects error rate, equalized odds, and statistical parity. While it clearly reduces error rate (Figure 14), abstaining has an unreliable effect on equalized odds (Figure 15) and statistical parity (Figure 16), similar to the behavior of the *Random* baseline. Figure 17 shows that the differences in variable distributions between the overall population and the set selected through the abstaining process are relatively small, except for the *Binomial NLL* distribution on the *Adult* dataset.

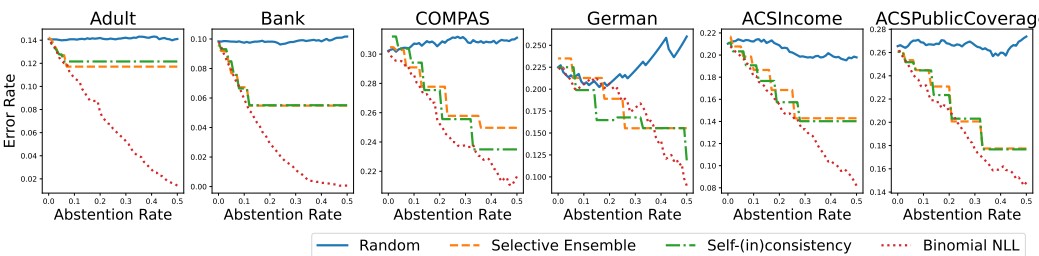

Figure 14: Abstention error rate.

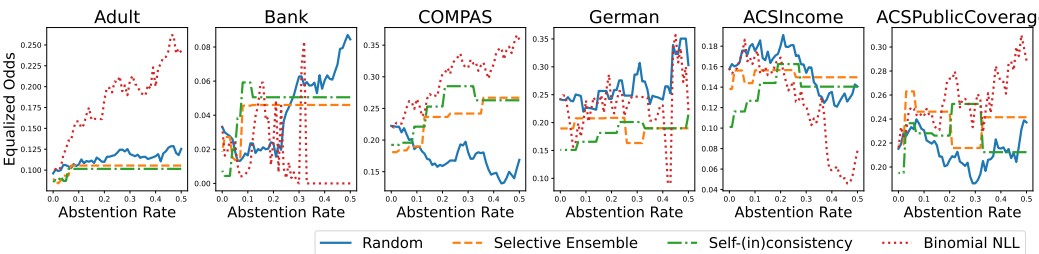

Figure 15: Abstention equalized odds.

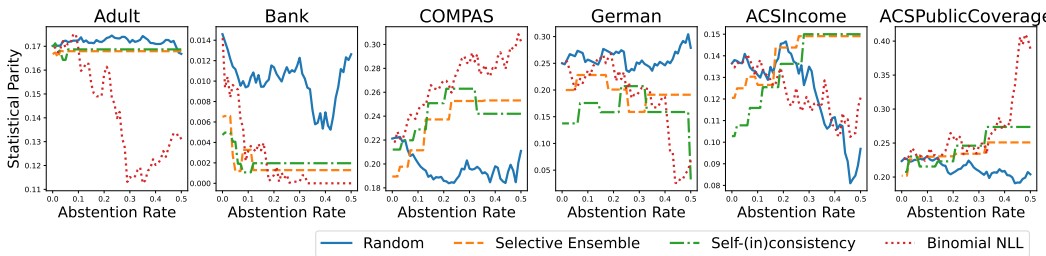

Figure 16: Abstention statistical parity.

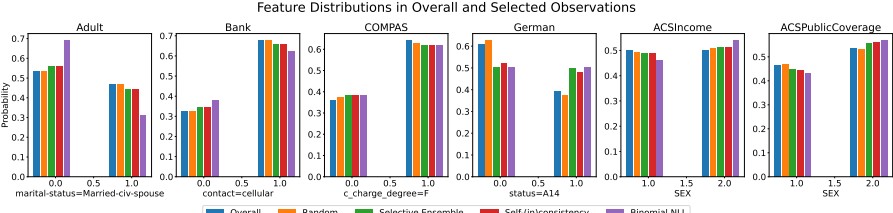

Figure 17: The variable with the largest difference (as measured by *Wasserstein* distance) between the distribution on the overall population and the set selected through the abstaining process. Except for the *Binomial NLL* distribution on the *Adult* dataset, the differences tend to be relatively small.

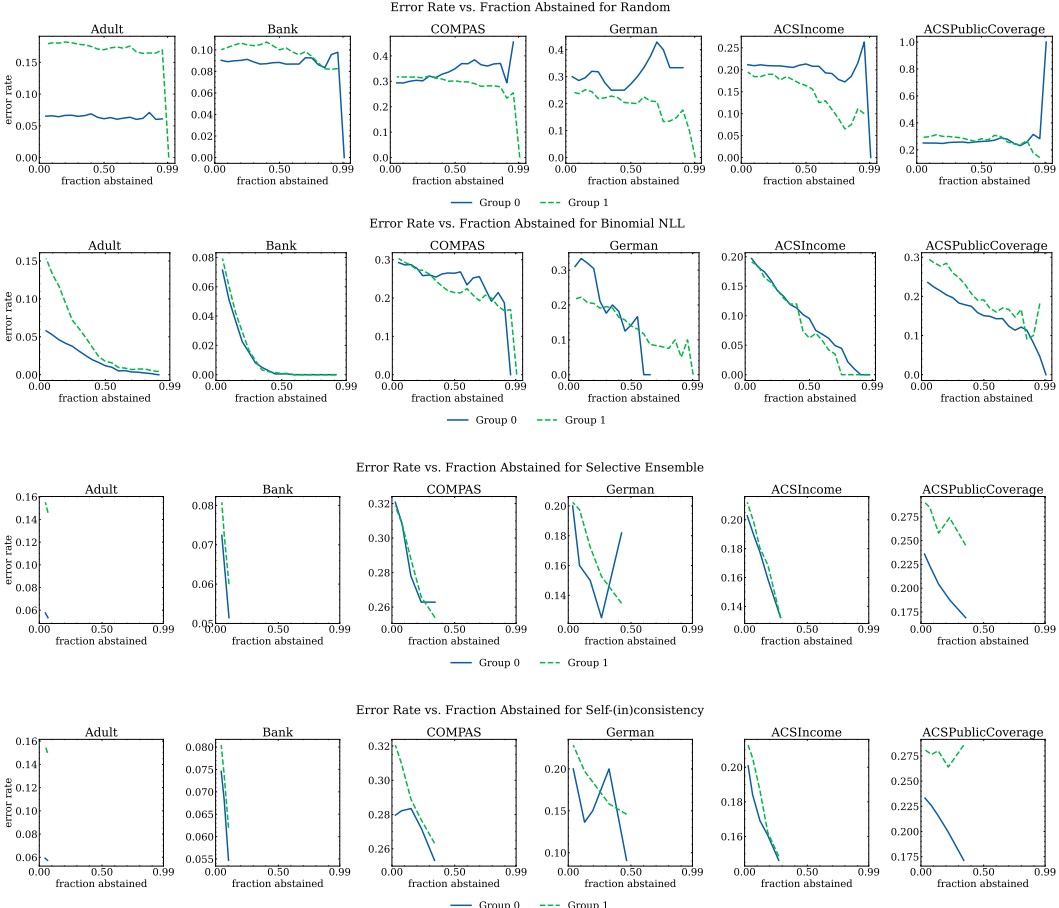

Figure 18: Error rate vs. abstained across different uncertainty algorithms. We set an overall abstention rate $r$ (shown on the $x$-axis) and plot the error rates for each protected group. Our results reveal that the algorithms yield varying error levels across protected groups on both the German and ACS Public Coverage datasets.

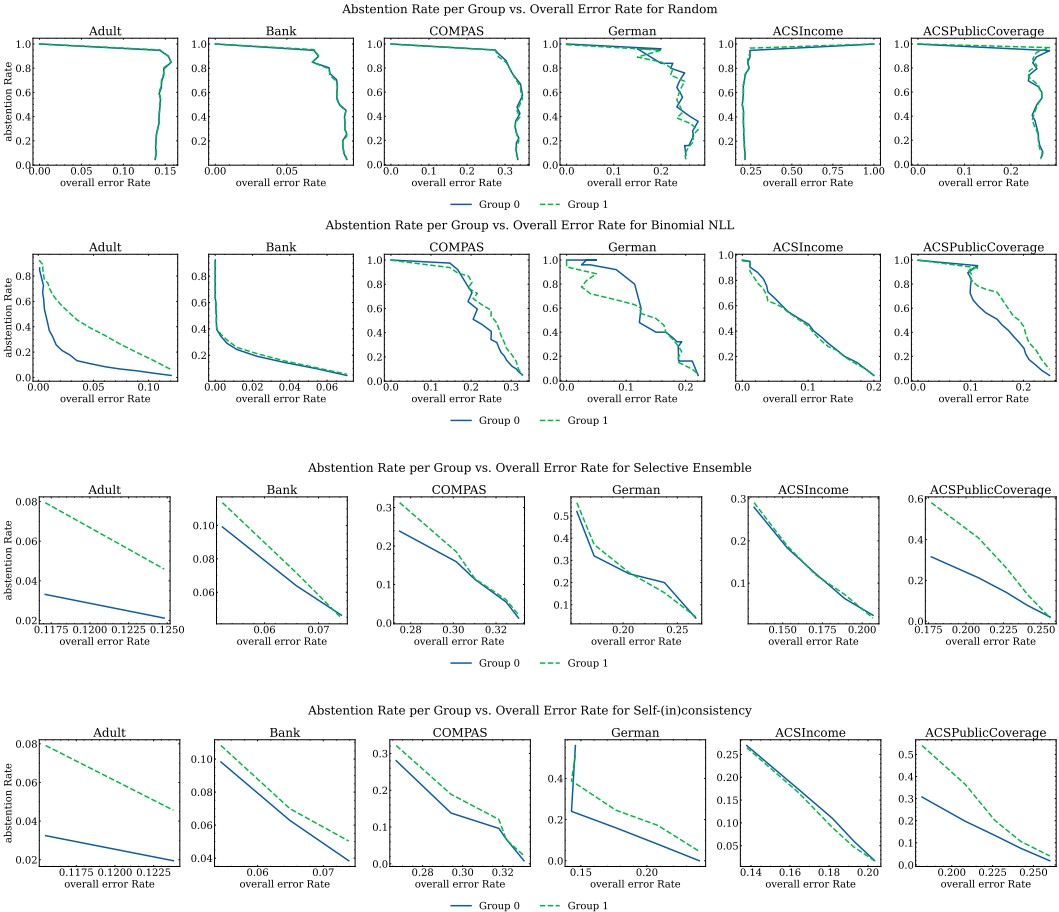

Figure 19: Abstention rate per group vs. overall error rate across different uncertainty algorithms. As expected, the random baseline exhibits no difference in abstention rates between protected groups, as its uncertainty estimates are random. In contrast, the Selective Ensemble demonstrates significant disparities in abstention rates across groups.

# F    BINARY FAIRNESS RESULTS

`FairlyUncertain` fairness benchmark on all five binary datasets. Abstaining reduces the error rate, and often equalized odds as a result, but has no improvement on statistical parity. Only *Predictive Parity* (the difference in the ratio of true positives to all positive labels assigned) was relatively low for abstaining methods, and is likely explained by none of the fairness methods explicitly optimizing this parameter.

| Approach | Error Rate | Statistical Parity | Equalized Odds | Equal Opportunity | Disparate Impact | Predictive Parity | False Positive Rate | Included % |
|---|---|---|---|---|---|---|---|---|
| Baseline | 0.22 ± 0.009 | 0.181 ± 0.023 | 0.165 ± 0.049 | 0.165 ± 0.049 | 1.65 ± 0.136 | 0.057 ± 0.034 | 0.083 ± 0.024 | 100.0 ± 0.0 |
| Threshold Optimizer SP | 0.233 ± 0.009 | 0.029 ± 0.021 | 0.135 ± 0.046 | 0.063 ± 0.033 | 0.979 ± 0.07 | 0.241 ± 0.043 | 0.134 ± 0.046 | 100.0 ± 0.0 |
| Threshold Optimizer EO | 0.226 ± 0.01 | 0.101 ± 0.03 | 0.08 ± 0.049 | 0.079 ± 0.049 | 1.3 ± 0.113 | 0.119 ± 0.039 | 0.024 ± 0.021 | 100.0 ± 0.0 |
| Exponentiated Gradient SP | 0.229 ± 0.011 | 0.041 ± 0.022 | 0.117 ± 0.038 | 0.065 ± 0.04 | 0.946 ± 0.08 | 0.203 ± 0.033 | 0.115 ± 0.038 | 100.0 ± 0.0 |
| Exponentiated Gradient EO | 0.218 ± 0.008 | 0.101 ± 0.041 | 0.092 ± 0.049 | 0.09 ± 0.052 | 1.3 ± 0.15 | 0.119 ± 0.045 | 0.031 ± 0.017 | 100.0 ± 0.0 |
| Grid Search SP | 0.23 ± 0.012 | 0.083 ± 0.045 | 0.142 ± 0.064 | 0.108 ± 0.067 | 1.04 ± 0.259 | 0.183 ± 0.048 | 0.106 ± 0.067 | 100.0 ± 0.0 |
| Grid Search EO | 0.227 ± 0.008 | 0.148 ± 0.034 | 0.132 ± 0.07 | 0.125 ± 0.074 | 1.48 ± 0.152 | 0.087 ± 0.048 | 0.056 ± 0.031 | 100.0 ± 0.0 |
| Random | 0.221 ± 0.009 | 0.174 ± 0.028 | 0.154 ± 0.046 | 0.15 ± 0.052 | 1.61 ± 0.155 | 0.044 ± 0.034 | 0.085 ± 0.024 | 88.3 ± 6.96 |
| Ensemble | 0.2 ± 0.019 | 0.193 ± 0.032 | 0.158 ± 0.063 | 0.155 ± 0.064 | 1.72 ± 0.18 | 0.032 ± 0.038 | 0.093 ± 0.031 | 89.0 ± 8.04 |
| Selective Ensemble | 0.201 ± 0.024 | 0.183 ± 0.03 | 0.154 ± 0.049 | 0.15 ± 0.059 | 1.69 ± 0.204 | 0.039 ± 0.037 | 0.082 ± 0.029 | 94.1 ± 7.29 |
| Self-(in)consistency | 0.188 ± 0.026 | 0.188 ± 0.028 | 0.155 ± 0.05 | 0.15 ± 0.058 | 1.73 ± 0.198 | 0.035 ± 0.028 | 0.081 ± 0.026 | 89.3 ± 8.87 |
| Binomial NLL | 0.176 ± 0.028 | 0.194 ± 0.021 | 0.153 ± 0.044 | 0.149 ± 0.051 | 1.77 ± 0.138 | 0.034 ± 0.032 | 0.08 ± 0.024 | 82.9 ± 9.5 |

Table 8: ACS Income. Note that this is reported in Table 3 of the main paper body to *2 significant digits* of precision. This was in order to make the font more legible - all results in this section are with 3 significant digits.

| Approach | Error Rate | Statistical Parity | Equalized Odds | Equal Opportunity | Disparate Impact | Predictive Parity | False Positive Rate | Included % |
|---|---|---|---|---|---|---|---|---|
| Baseline | 0.265 ± 0.011 | 0.25 ± 0.031 | 0.211 ± 0.04 | 0.168 ± 0.061 | 0.479 ± 0.037 | 0.031 ± 0.03 | 0.194 ± 0.041 | 100.0 ± 0.0 |
| Threshold Optimizer SP | 0.291 ± 0.016 | 0.038 ± 0.025 | 0.14 ± 0.033 | 0.129 ± 0.046 | 0.882 ± 0.07 | 0.034 ± 0.038 | 0.077 ± 0.031 | 100.0 ± 0.0 |
| Threshold Optimizer EO | 0.288 ± 0.017 | 0.168 ± 0.028 | 0.141 ± 0.037 | 0.069 ± 0.045 | 0.637 ± 0.042 | 0.064 ± 0.039 | 0.139 ± 0.04 | 100.0 ± 0.0 |
| Exponentiated Gradient SP | 0.279 ± 0.008 | 0.035 ± 0.021 | 0.091 ± 0.052 | 0.088 ± 0.055 | 0.961 ± 0.116 | 0.152 ± 0.039 | 0.032 ± 0.02 | 100.0 ± 0.0 |
| Exponentiated Gradient EO | 0.277 ± 0.014 | 0.176 ± 0.036 | 0.148 ± 0.034 | 0.073 ± 0.048 | 0.62 ± 0.053 | 0.051 ± 0.04 | 0.147 ± 0.035 | 100.0 ± 0.0 |
| Grid Search SP | 0.291 ± 0.013 | 0.077 ± 0.023 | 0.124 ± 0.07 | 0.115 ± 0.079 | 1.01 ± 0.255 | 0.139 ± 0.08 | 0.072 ± 0.042 | 100.0 ± 0.0 |
| Grid Search EO | 0.276 ± 0.014 | 0.221 ± 0.034 | 0.184 ± 0.037 | 0.12 ± 0.054 | 0.557 ± 0.043 | 0.041 ± 0.046 | 0.181 ± 0.041 | 100.0 ± 0.0 |
| Random | 0.263 ± 0.012 | 0.239 ± 0.041 | 0.205 ± 0.042 | 0.152 ± 0.077 | 0.495 ± 0.049 | 0.028 ± 0.042 | 0.185 ± 0.043 | 88.4 ± 8.66 |
| Ensemble | 0.248 ± 0.019 | 0.281 ± 0.044 | 0.225 ± 0.054 | 0.21 ± 0.052 | 0.42 ± 0.036 | 0.015 ± 0.027 | 0.191 ± 0.042 | 89.6 ± 7.63 |
| Selective Ensemble | 0.241 ± 0.02 | 0.271 ± 0.033 | 0.236 ± 0.05 | 0.211 ± 0.073 | 0.412 ± 0.048 | 0.024 ± 0.022 | 0.169 ± 0.044 | 89.0 ± 8.52 |
| Self-(in)consistency | 0.251 ± 0.02 | 0.249 ± 0.04 | 0.202 ± 0.055 | 0.183 ± 0.064 | 0.447 ± 0.045 | 0.029 ± 0.025 | 0.162 ± 0.051 | 93.3 ± 7.86 |
| Binomial NLL | 0.236 ± 0.024 | 0.257 ± 0.034 | 0.207 ± 0.056 | 0.175 ± 0.071 | 0.436 ± 0.037 | 0.012 ± 0.021 | 0.166 ± 0.05 | 87.0 ± 7.32 |

Table 9: ACS Public Coverage.

| Approach | Error Rate | Statistical Parity | Equalized Odds | Equal Opportunity | Disparate Impact | Predictive Parity | False Positive Rate | Included % |
|---|---|---|---|---|---|---|---|---|
| Baseline | 0.139 ± 0.002 | 0.168 ± 0.006 | 0.092 ± 0.022 | 0.092 ± 0.022 | 0.291 ± 0.017 | 0.037 ± 0.017 | 0.058 ± 0.005 | 100.0 ± 0.0 |
| Threshold Optimizer SP | 0.185 ± 0.002 | 0.007 ± 0.007 | 0.132 ± 0.01 | 0.024 ± 0.015 | 0.999 ± 0.037 | 0.499 ± 0.013 | 0.132 ± 0.01 | 100.0 ± 0.0 |
| Threshold Optimizer EO | 0.15 ± 0.002 | 0.122 ± 0.008 | 0.026 ± 0.018 | 0.024 ± 0.019 | 0.53 ± 0.028 | 0.271 ± 0.03 | 0.01 ± 0.006 | 100.0 ± 0.0 |
| Exponentiated Gradient SP | 0.15 ± 0.004 | 0.019 ± 0.009 | 0.288 ± 0.023 | 0.288 ± 0.023 | 0.903 ± 0.045 | 0.335 ± 0.019 | 0.054 ± 0.006 | 100.0 ± 0.0 |
| Exponentiated Gradient EO | 0.135 ± 0.002 | 0.134 ± 0.005 | 0.034 ± 0.008 | 0.028 ± 0.012 | 0.433 ± 0.017 | 0.11 ± 0.032 | 0.03 ± 0.006 | 100.0 ± 0.0 |
| Grid Search SP | 0.162 ± 0.004 | 0.029 ± 0.009 | 0.359 ± 0.013 | 0.359 ± 0.013 | 1.17 ± 0.055 | 0.412 ± 0.021 | 0.091 ± 0.009 | 100.0 ± 0.0 |
| Grid Search EO | 0.132 ± 0.001 | 0.18 ± 0.005 | 0.075 ± 0.007 | 0.074 ± 0.009 | 0.318 ± 0.008 | 0.018 ± 0.018 | 0.062 ± 0.005 | 100.0 ± 0.0 |
| Random | 0.139 ± 0.002 | 0.168 ± 0.006 | 0.087 ± 0.018 | 0.087 ± 0.018 | 0.293 ± 0.017 | 0.032 ± 0.019 | 0.058 ± 0.005 | 87.1 ± 6.8 |
| Ensemble | 0.115 ± 0.023 | 0.171 ± 0.01 | 0.09 ± 0.028 | 0.09 ± 0.029 | 0.24 ± 0.047 | 0.03 ± 0.019 | 0.046 ± 0.009 | 86.6 ± 11.6 |
| Selective Ensemble | 0.129 ± 0.008 | 0.165 ± 0.008 | 0.105 ± 0.026 | 0.105 ± 0.026 | 0.271 ± 0.022 | 0.032 ± 0.021 | 0.049 ± 0.008 | 96.4 ± 2.54 |
| Self-(in)consistency | 0.134 ± 0.008 | 0.166 ± 0.007 | 0.101 ± 0.029 | 0.101 ± 0.029 | 0.276 ± 0.025 | 0.037 ± 0.021 | 0.052 ± 0.006 | 97.9 ± 2.21 |
| Binomial NLL | 0.106 ± 0.025 | 0.153 ± 0.007 | 0.122 ± 0.053 | 0.122 ± 0.053 | 0.263 ± 0.042 | 0.016 ± 0.01 | 0.033 ± 0.015 | 89.6 ± 7.89 |

Table 10: Adult.

| Approach | Error Rate | Statistical Parity | Equalized Odds | Equal Opportunity | Disparate Impact | Predictive Parity | False Positive Rate | Included % |
|---|---|---|---|---|---|---|---|---|
| Baseline | 0.092 ± 0.002 | 0.012 ± 0.006 | 0.053 ± 0.027 | 0.053 ± 0.027 | 0.925 ± 0.096 | 0.033 ± 0.016 | 0.004 ± 0.004 | 100.0 ± 0.0 |
| Threshold Optimizer SP | 0.096 ± 0.002 | 0.006 ± 0.003 | 0.023 ± 0.014 | 0.023 ± 0.015 | 1.01 ± 0.062 | 0.038 ± 0.022 | 0.005 ± 0.003 | 100.0 ± 0.0 |
| Threshold Optimizer EO | 0.097 ± 0.002 | 0.006 ± 0.004 | 0.025 ± 0.01 | 0.025 ± 0.011 | 0.975 ± 0.06 | 0.02 ± 0.018 | 0.003 ± 0.002 | 100.0 ± 0.0 |
| Exponentiated Gradient SP | 0.096 ± 0.002 | 0.006 ± 0.004 | 0.032 ± 0.017 | 0.032 ± 0.017 | 0.986 ± 0.067 | 0.035 ± 0.025 | 0.004 ± 0.003 | 100.0 ± 0.0 |
| Exponentiated Gradient EO | 0.096 ± 0.002 | 0.005 ± 0.003 | 0.026 ± 0.015 | 0.026 ± 0.016 | 0.99 ± 0.05 | 0.039 ± 0.025 | 0.005 ± 0.002 | 100.0 ± 0.0 |
| Grid Search SP | 0.097 ± 0.002 | 0.009 ± 0.006 | 0.038 ± 0.021 | 0.038 ± 0.021 | 0.93 ± 0.056 | 0.031 ± 0.031 | 0.004 ± 0.004 | 100.0 ± 0.0 |
| Grid Search EO | 0.097 ± 0.002 | 0.009 ± 0.006 | 0.038 ± 0.021 | 0.038 ± 0.021 | 0.93 ± 0.056 | 0.031 ± 0.031 | 0.004 ± 0.004 | 100.0 ± 0.0 |
| Random | 0.092 ± 0.002 | 0.01 ± 0.005 | 0.049 ± 0.028 | 0.048 ± 0.029 | 0.934 ± 0.084 | 0.031 ± 0.017 | 0.003 ± 0.003 | 91.0 ± 7.38 |
| Ensemble | 0.045 ± 0.026 | 0.005 ± 0.004 | 0.067 ± 0.05 | 0.067 ± 0.05 | 0.985 ± 0.135 | 0.021 ± 0.019 | 0.001 ± 0.0 | 84.3 ± 8.45 |
| Selective Ensemble | 0.077 ± 0.013 | 0.007 ± 0.004 | 0.043 ± 0.035 | 0.042 ± 0.036 | 0.922 ± 0.071 | 0.033 ± 0.021 | 0.002 ± 0.002 | 95.7 ± 3.8 |
| Self-(in)consistency | 0.076 ± 0.013 | 0.008 ± 0.007 | 0.053 ± 0.044 | 0.053 ± 0.045 | 0.902 ± 0.109 | 0.038 ± 0.032 | 0.002 ± 0.002 | 95.8 ± 3.82 |
| Binomial NLL | 0.067 ± 0.028 | 0.007 ± 0.005 | 0.043 ± 0.031 | 0.043 ± 0.032 | 0.908 ± 0.074 | 0.033 ± 0.029 | 0.001 ± 0.001 | 92.2 ± 8.74 |

Table 11: Bank.

| Approach | Error Rate | Statistical Parity | Equalized Odds | Equal Opportunity | Disparate Impact | Predictive Parity | False Positive Rate | Included % |
|---|---|---|---|---|---|---|---|---|
| Baseline | 0.316 ± 0.012 | 0.194 ± 0.027 | 0.188 ± 0.05 | 0.183 ± 0.054 | 0.542 ± 0.055 | 0.075 ± 0.051 | 0.125 ± 0.039 | 100.0 ± 0.0 |
| Threshold Optimizer SP | 0.344 ± 0.015 | 0.042 ± 0.025 | 0.112 ± 0.047 | 0.074 ± 0.052 | 0.963 ± 0.107 | 0.227 ± 0.067 | 0.09 ± 0.053 | 100.0 ± 0.0 |
| Threshold Optimizer EO | 0.35 ± 0.016 | 0.065 ± 0.031 | 0.074 ± 0.048 | 0.066 ± 0.052 | 0.82 ± 0.084 | 0.163 ± 0.067 | 0.04 ± 0.028 | 100.0 ± 0.0 |
| Exponentiated Gradient SP | 0.329 ± 0.013 | 0.041 ± 0.025 | 0.051 ± 0.027 | 0.038 ± 0.033 | 0.902 ± 0.069 | 0.161 ± 0.05 | 0.028 ± 0.018 | 100.0 ± 0.0 |
| Exponentiated Gradient EO | 0.332 ± 0.014 | 0.083 ± 0.028 | 0.085 ± 0.046 | 0.084 ± 0.047 | 0.792 ± 0.071 | 0.164 ± 0.066 | 0.038 ± 0.025 | 100.0 ± 0.0 |
| Grid Search SP | 0.327 ± 0.015 | 0.047 ± 0.058 | 0.112 ± 0.06 | 0.097 ± 0.063 | 0.991 ± 0.182 | 0.176 ± 0.043 | 0.07 ± 0.042 | 100.0 ± 0.0 |
| Grid Search EO | 0.323 ± 0.012 | 0.213 ± 0.03 | 0.224 ± 0.059 | 0.223 ± 0.06 | 0.517 ± 0.054 | 0.088 ± 0.063 | 0.135 ± 0.034 | 100.0 ± 0.0 |
| Random | 0.317 ± 0.014 | 0.181 ± 0.025 | 0.168 ± 0.04 | 0.154 ± 0.052 | 0.573 ± 0.051 | 0.063 ± 0.067 | 0.12 ± 0.043 | 86.3 ± 6.72 |
| Ensemble | 0.305 ± 0.011 | 0.201 ± 0.033 | 0.206 ± 0.06 | 0.197 ± 0.076 | 0.5 ± 0.09 | 0.056 ± 0.048 | 0.127 ± 0.022 | 87.8 ± 8.68 |
| Selective Ensemble | 0.288 ± 0.015 | 0.211 ± 0.047 | 0.213 ± 0.07 | 0.202 ± 0.088 | 0.498 ± 0.107 | 0.054 ± 0.046 | 0.126 ± 0.032 | 88.0 ± 7.52 |
| Self-(in)consistency | 0.287 ± 0.019 | 0.213 ± 0.033 | 0.219 ± 0.071 | 0.216 ± 0.074 | 0.485 ± 0.067 | 0.07 ± 0.06 | 0.121 ± 0.034 | 86.9 ± 9.61 |
| Binomial NLL | 0.292 ± 0.018 | 0.206 ± 0.033 | 0.199 ± 0.058 | 0.196 ± 0.061 | 0.5 ± 0.074 | 0.052 ± 0.047 | 0.123 ± 0.035 | 88.7 ± 8.4 |

Table 12: COMPAS.

| Approach | Error Rate | Statistical Parity | Equalized Odds | Equal Opportunity | Disparate Impact | Predictive Parity | False Positive Rate | Included % |
|---|---|---|---|---|---|---|---|---|
| Baseline | 0.251 ± 0.025 | 0.129 ± 0.06 | 0.147 ± 0.064 | 0.129 ± 0.07 | 1.72 ± 0.35 | 0.082 ± 0.048 | 0.094 ± 0.067 | 100.0 ± 0.0 |
| Threshold Optimizer SP | 0.258 ± 0.026 | 0.081 ± 0.059 | 0.175 ± 0.082 | 0.152 ± 0.097 | 1.08 ± 0.469 | 0.187 ± 0.11 | 0.078 ± 0.062 | 100.0 ± 0.0 |
| Threshold Optimizer EO | 0.255 ± 0.029 | 0.155 ± 0.081 | 0.166 ± 0.097 | 0.151 ± 0.065 | 1.75 ± 0.423 | 0.105 ± 0.105 | 0.124 ± 0.092 | 100.0 ± 0.0 |
| Exponentiated Gradient SP | 0.264 ± 0.025 | 0.065 ± 0.056 | 0.11 ± 0.066 | 0.089 ± 0.063 | 1.29 ± 0.371 | 0.112 ± 0.112 | 0.063 ± 0.068 | 100.0 ± 0.0 |
| Exponentiated Gradient EO | 0.255 ± 0.029 | 0.155 ± 0.081 | 0.166 ± 0.097 | 0.151 ± 0.065 | 1.75 ± 0.423 | 0.105 ± 0.105 | 0.124 ± 0.092 | 100.0 ± 0.0 |
| Grid Search SP | 0.269 ± 0.023 | 0.136 ± 0.079 | 0.175 ± 0.073 | 0.16 ± 0.072 | 1.46 ± 0.602 | 0.086 ± 0.068 | 0.111 ± 0.078 | 100.0 ± 0.0 |
| Grid Search EO | 0.259 ± 0.038 | 0.15 ± 0.094 | 0.197 ± 0.108 | 0.186 ± 0.109 | 1.67 ± 0.443 | 0.109 ± 0.1 | 0.108 ± 0.073 | 100.0 ± 0.0 |
| Random | 0.25 ± 0.026 | 0.116 ± 0.057 | 0.139 ± 0.075 | 0.107 ± 0.091 | 1.64 ± 0.315 | 0.068 ± 0.04 | 0.087 ± 0.06 | 92.5 ± 8.25 |
| Ensemble | 0.222 ± 0.046 | 0.125 ± 0.076 | 0.155 ± 0.088 | 0.142 ± 0.1 | 2.04 ± 0.753 | 0.069 ± 0.077 | 0.077 ± 0.054 | 85.4 ± 9.95 |
| Selective Ensemble | 0.214 ± 0.045 | 0.122 ± 0.064 | 0.176 ± 0.097 | 0.157 ± 0.113 | 1.93 ± 0.496 | 0.082 ± 0.084 | 0.073 ± 0.047 | 88.4 ± 8.57 |
| Self-(in)consistency | 0.216 ± 0.046 | 0.109 ± 0.063 | 0.148 ± 0.084 | 0.142 ± 0.092 | 1.85 ± 0.481 | 0.068 ± 0.087 | 0.062 ± 0.042 | 90.3 ± 8.45 |
| Binomial NLL | 0.232 ± 0.023 | 0.107 ± 0.049 | 0.143 ± 0.085 | 0.132 ± 0.091 | 1.68 ± 0.351 | 0.057 ± 0.034 | 0.065 ± 0.054 | 92.6 ± 7.86 |

Table 13: German.

# G  BINARY CALIBRATION WITH DIFFERENT SIZED GROUPS

We find that bigger groups (a smaller number of groups) exhibit more identifiable patterns with a positive correlation whereas smaller groups (a bigger number of groups) exhibit slightly more noisy behavior. These results were generated using an underlying XGBoost model.

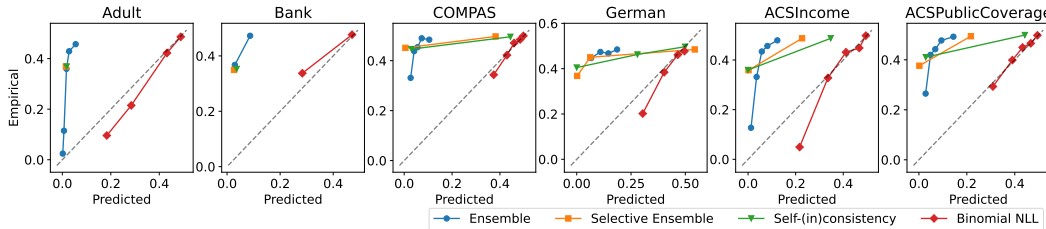

Figure 20: For five groups assembled by predicted uncertainty, we plot the average predicted uncertainty against the empirical standard deviation of the outcomes.

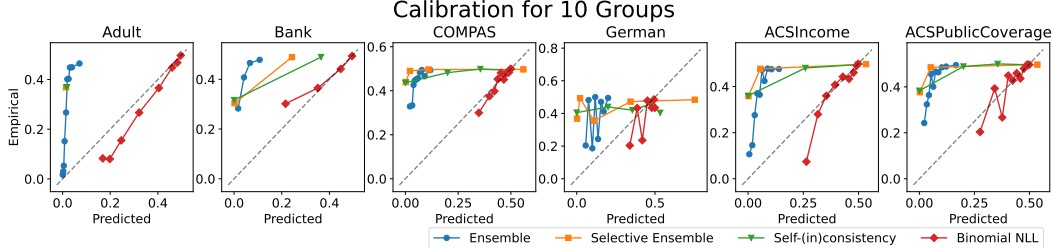

Figure 21: For ten groups assembled by predicted uncertainty, we plot the average predicted uncertainty against the empirical standard deviation of the outcomes.

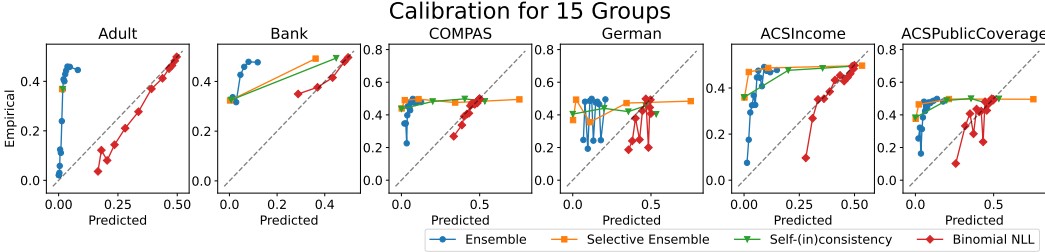

Figure 22: For fifteen groups assembled by predicted uncertainty, we plot the average predicted uncertainty against the empirical standard deviation of the outcomes.

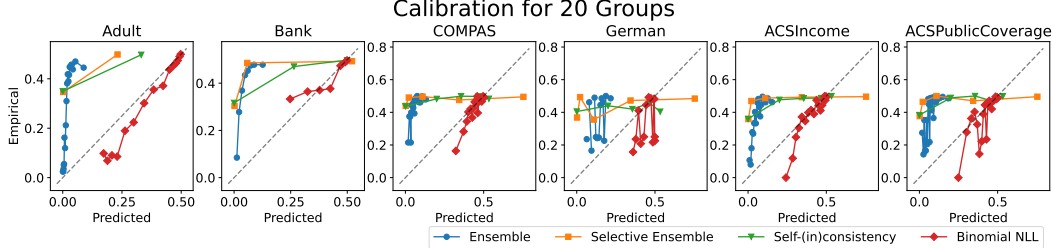

Figure 23: For twenty groups assembled by predicted uncertainty, we plot the average predicted uncertainty against the empirical standard deviation of the outcomes.

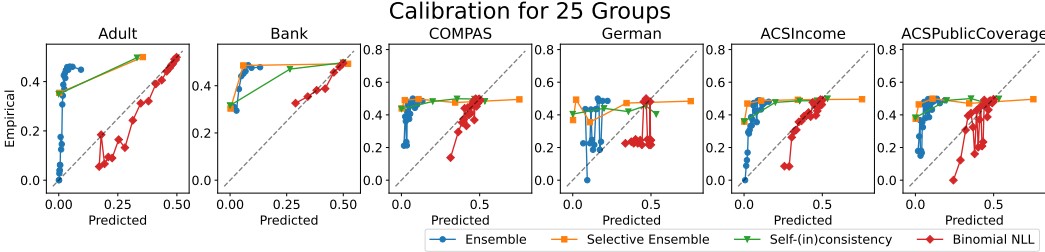

Figure 24: For twenty five groups assembled by predicted uncertainty, we plot the average predicted uncertainty against the empirical standard deviation of the outcomes.

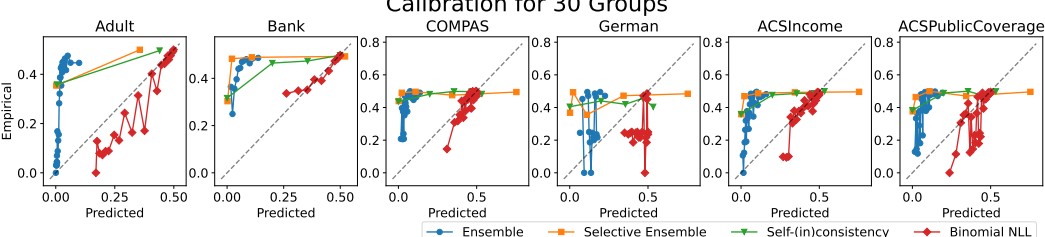

Figure 25: For thirty groups assembled by predicted uncertainty, we plot the average predicted uncertainty against the empirical standard deviation of the outcomes.

## H  NOTES ON XGBOOST AND CUSTOMIZING THE LOSS

XGBoost optimizes a loss using its gradient $g_i$ for first-order information and its Hessian $h_i$ for second-order information in a relatively standard, higher order optimization framework (see Section 2.2. in Chen & Guestrin (2016)). Intuitively, $g_i$ indicates the direction for improving model predictions, and $h_i$ is useful for determining the curvature or "rate of change" of the loss. During XGBoost tree construction, each tree is built iteratively in order to correct the "residuals" left by previous trees. Here, "residuals" refer to the differences between the observed values of the target variable and the values predicted by the model e.g. the errors in the predictions made by the model. We let

$$w_j^* = -\frac{\sum_{i \in I_j} g_i}{\sum_{i \in I_j} h_i + \lambda}, \tag{1}$$

where $w_j^*$ is the optimal weight for the $j$-th leaf, $I_j$ represents the set of instances in leaf $j$, and $\lambda$ is a regularizer. Formulating it in this way is the classic balancing act - we want to reduce the loss, but we regularize against the complexity of the model. Then, when we want to evaluate potential splits during tree construction, the "gain" from a split is calculated as:

$$\text{Gain} = \frac{1}{2} \left[ \frac{(\sum_{i \in I_L} g_i)^2}{\sum_{i \in I_L} h_i + \lambda} + \frac{(\sum_{i \in I_R} g_i)^2}{\sum_{i \in I_R} h_i + \lambda} - \frac{(\sum_{i \in I} g_i)^2}{\sum_{i \in I} h_i + \lambda} \right] - \gamma, \tag{2}$$

where $I_L$ and $I_R$ are the instance sets of the left and right child nodes post-split. Thus, specifying a custom loss for XGBoost simply requires deriving the first order gradient and second order hessian for a given loss function with arbitrary outputs.

# I FAIRNESS METRICS

Below we provide formulas for the fairness metrics included in our benchmark. Let $X$ denote the set of features, $Y$ the true label, $\hat{Y}$ the predicted label, and $A$ the protected attribute (e.g., race, gender). Note that we generally report these as *metrics*, which is to say we quantify the degree of fairness by calculating the absolute difference between the two sides of the equality in each definition. For disparate impact, we report $\delta$ directly.

**Definition I.1** (Statistical Parity).

$$P(\hat{Y} = 1 \mid A = a) = P(\hat{Y} = 1 \mid A = a') \quad \forall a, a' \tag{3}$$

*i.e. the probability of receiving a positive outcome should be the same across all $A$.*

**Definition I.2** (Equalized Odds).

$$P(\hat{Y} = 1 \mid A = a, Y = y) = P(\hat{Y} = 1 \mid A = a', Y = y) \quad \forall a, a', y \tag{4}$$

*i.e. the prediction outcome is conditionally independent of the protected attribute $A$ given $Y$.*

**Definition I.3** (Equal Opportunity).

$$P(\hat{Y} = 1 \mid A = a, Y = 1) = P(\hat{Y} = 1 \mid A = a', Y = 1) \quad \forall a, a' \tag{5}$$

*i.e. the true positive rate is the same across all $A$.*

**Definition I.4** (Disparate Impact).

$$\frac{P(\hat{Y} = 1 \mid A = a)}{P(\hat{Y} = 1 \mid A = a')} \geq \delta \quad \textit{for some threshold } \delta \tag{6}$$

*i.e. the ratio of positive outcomes between groups remains close to 1 (typically $\delta = 0.8$).*

**Definition I.5** (Predictive Parity).

$$P(Y = 1 \mid \hat{Y} = 1, A = a) = P(Y = 1 \mid \hat{Y} = 1, A = a') \quad \forall a, a' \tag{7}$$

*i.e. precision (the probability of a true positive given a positive prediction) is the same across all $A$.*

**Definition I.6** (False Positive Rate Equality).

$$P(\hat{Y} = 1 \mid A = a, Y = 0) = P(\hat{Y} = 1 \mid A = a', Y = 0) \quad \forall a, a' \tag{8}$$

*i.e. the false positive rate is the same across all $A$.*

