# OpenReview forum: "FairlyUncertain: A Comprehensive Benchmark of Uncertainty in Algorithmic Fairness"
_ICLR.cc/2025/Conference — Submitted to ICLR 2025_

### Official Review · Reviewer_ufZ7 · 2024-11-01

**Soundness:** 2
**Presentation:** 3
**Contribution:** 3
**Rating:** 5
**Confidence:** 4

**Summary:**

This paper presents the FairlyUncertain benchmark, which is aimed at measuring the fairness properties of model uncertainty estimates. They focus on two metrics which are called consistency and calibration, and explore the results on these metrics across several algorithms and several datasets, concluding that probabilistically-grounded methods tend to return the best results. They explore the ramifications of abstention on fairness metrics, and propose a new uncertainty-aware form of statistical parity.

**Strengths:**

- this is an important topic for which there are no currently existing benchmarks that I’m aware of, I think this could be useful for people
- good level of thoroughness here exploring the various different metrics across datasets
- abstention experiments are interesting - surprising to me that abstention doesn’t improve most fairness metrics
- uncertainty-aware Statistical Parity is a good idea - incorporating the uncertainty estimate into a CDF-based metric makes sense

**Weaknesses:**

- I’m not sure I agree with the characterization (around L90) of A/B types as unmeasurable and C/D/E types as estimatable from a fixed sample. I think both of these statements are conditional on various distributional assumptions - for instance, you can measure individual variance if you assume you know the mean (or its parametric form and inputs). I think explicating the underlying data generating process would make this whole contribution more substantial
- A couple issues with formalisms in Section 2: 1. I don’t understand Def 2.2 - “differ only in hyperparameter settings” and “there is some j…” seem to say different things: the first implies P and P’ are in some shared “pipeline class” parametrized by different hyperparameters, the second implies that just one hyperparameter differs. 2. On Line 127, f(P) is used - this seems wrong to me, as on L117, f is named as the *output* of P, not a function of it
- I’m not sure I agree with the given definition of consistency as a useful fairness axiom - it seems like more of a Lipschitz condition over what a well-parametrized pipeline might look like. My impression of a fairness axiom of this type might be of the flavor that uncertainty estimates should be as invariant as possible to hyperparameter choices (I’m not sure this is a good axiom either but it seems more fairness-related).
- there is a rich literature on calibration metrics which is not connected to at all in this paper - it would be good to get a better understanding of how those relate to the chosen metrics
- L287 and elsewhere: for consistency, it seems like a non-robust choice to output a maximum std. Dev over individuals - I’m wondering if like a 90th percentile might be a better choice
- L391 - it seems odd to optimize an objective function that mixes statistical parity and equalized odds. Usually, one or the other is picked, since they are very different objectives
- Def 6.2 - it seems overly specific to me to assume that the model outputs are parametrizing a normal distribution. In particular, I don’t think this makes sense in the binary Y case - and the end of the definition (CDF comparison for all outcomes y \in Y) I think doesn’t make sense in the binary y case either (rather you want this to be true at all points in the range)

Small points:
- L142: “a model that always returns 0 is consistent” - by the previous definition this is not true, as the definition of consistency applies to a pipeline class, rather than the model which is outputted. Such a pipeline may output a constant-0 model for some set of hyperparameters, but not others.
- how is clustering done for calibration? I don’t think this is discussed
- Table 4: not sure how Disp. Impact is defined

**Questions:**

- I’m not sure the framework in Table 1 does a ton for me - doesn’t really connect to the rest of the paper at the moment so tightly and I’m not sure what the takeaways are. I would be interested to see a clearer explication of how these types of uncertainty connect to epistemic/aleatoric
- would like to see a more fleshed out argument for the consistency axiom
- would be interested to know more about connections to current calibration literature
- how much do results change if each fairness metric is optimized for individually (in combination with error)
- is there a more general version of the UA-SP definition?

---

> ### Author Response · Authors · 2024-11-21
>
> Thank you for your thorough review and the suggestions to enhance our paper! We will respond to each of your comments/questions below, briefly summarizing the comment inline.
>
> > **disagree with characterization of A/B types as unmeasurable and C/D/E types as estimatable from a fixed sample…explicating the underlying data generating process would make this whole contribution more substantial…interested to see a clearer explication of how these types of uncertainty connect to epistemic/aleatoric**
>
> W1/Q1. Good point! We agree this characterization of uncertainty is not tied to the paper, and does not contribute much. As such, we’ve moved it to the appendix. Our original intention was to provide a more nuanced taxonomy of aleatoric and epistemic uncertainty.
>
> >  **Def 2.2 - “differ only in hyperparameter settings” and “there is some j…” seem to say different things: the first implies P and P’ are in some shared “pipeline class” parametrized by different hyperparameters, the second implies that just one hyperparameter differs**
>
> W2. Good point! In defining “similar learning pipelines”, we are walking the line between generality and formalization. Our intention is to define $P$ and $P’$ in some shared “pipeline class”, but then we need to define what this pipeline class means. Since hyperparameter values are on different scales (e.g., learning rate and number of layers), it becomes difficult to talk about $P$ and $P’$ that differ in multiple hyperparameters but are still “close”. Hence, we formalize the pipeline class as $P$ and $P’$ that differ only in a single variable, at the cost of generality. This is a tradeoff we are willing to accept because, from a practical perspective, we use consistency as a one-sided test. As such, we would rather have a formal test that, if an algorithm fails it, tells us that the algorithm is not consistent rather than a more general test that is difficult to implement. That said, we welcome discussion of another formal definition of pipeline class.
>
> > **On Line 127, f(P) is used - this seems wrong to me, as on L117, f is named as the output of P, not a function of it**
>
> W2. Yes, the notation $f(P)$ to describe a trained model created from a learning pipeline $P$ is strange; we have updated it to $f_P$ to indicate that the model $f$ was produced by learning pipeline $P$.
>
> > **I’m not sure I agree with the given definition of consistency as a useful fairness axiom - it seems like more of a Lipschitz condition over what a well-parametrized pipeline might look like. My impression of a fairness axiom of this type might be of the flavor that uncertainty estimates should be as invariant as possible to hyperparameter choices…more fleshed out argument for the consistency axiom.**
>
> W3/Q2. The intention of the consistency axiom is to be invariant to hyperparameter choices, as you suggest. The challenge is that clearly the predictions have to have some dependence on hyperparameters (otherwise the hyperparameters don’t matter). Hence we chose to formalize invariance as requiring that small changes in hyperparameters do not substantially change the predictions. We chose this formalization because it is the closest to the invariance goal that we have come up with, and we would very much welcome another formalization that more closely aligns with invariance.
>
> > **there is a rich literature on calibration metrics which is not connected to at all in this paper - it would be good to get a better understanding of how those relate to the chosen metrics / would be interested to know more about connections to current calibration literature**
>
> W4/Q3. The literature that we are familiar with on calibration metrics broadly fall into either:
>
> 1. Expected Calibration Error (ECE): This approach buckets observations by the uncertainty predictions and computes the difference between the average predicted uncertainty and the average observed uncertainty. Instead of reporting this as a metric (e.g., a weighted average), we plot the predicted versus observed uncertainty in our “qualitative” calibration experiments.
>
> 2. Negative Log Likelihood (NLL): This approach makes an assumption on the underlying distribution and measures how likely we are to observe the outcomes that we did if the predicted parameters of the assumed distribution were correct. We report this in our “quantitative” calibration experiments.
>
> **We have added discussion on ECE vs. NLL in Appendix Section D.1 in the revised PDF, and have reported calibration metrics for ECE in Table 7, also in the Appendix (these results confirm our results on the NLL calibration metric).** Additionally, in the final version of the paper, we would be happy to discuss more literature on calibration metrics, especially in relation to our work. Please let us know if there are additional calibration metrics that you think we should report (beyond the ECE plots and the NLL values).

---

> > ### Comment · Reviewer_ufZ7 · 2024-11-26
> > **Response**
> >
> > Thanks for the response.
> >
> > Def 2.2. - I think this may just be a clarity question - the language says "differ only in hyperparameter settings" which seems to indicate that they may different in multiple hyperparameters, but the formalization suggests only 1 hyperparameter will differ
> >
> > Def of consistency as a fairness axiom: the new text connecting consistency to fairness principles is helpful. I still find it a bit odd in the sense that Lipschitzness and invariance are somewhat separate goals - for instance, a Lipschitz function can still respond quite wildly to large changes in input. I would instead think of a natural objective as being, for instance, bounded change in output for any change in input. I also wonder how these definitions correspond to various reparameterizations of the hyperparameter space. In any case, I do think invariance to hyperparameters is a reasonable thing to want but I'm not totally convinced that a) this definition achieves that, or b) it's super fairness related (although the added argument helps)
> >
> > Calibration metrics: ECE is a fine metric to use in the table, I don't think others are necessary - I do think there's more of a literature out there that might be work looking at in the related work (eg "Measuring Calibration in Deep Learning" is an example paper from this literature)
> >
> > I think there are useful pieces of this paper but I'm not sure I'm ready to raise my score here. I will take another look at the other reviews as well.

---

> > > ### Author Response · Authors · 2024-11-26
> > >
> > > > **Def 2.2. - I think this may just be a clarity question…indicate that they may different in multiple hyperparameters,**
> > >
> > > Ah, this is a good catch, thank you - we will change the definition such that it says:
> > >
> > > “Two learning pipelines $\mathcal{P}$ and $\mathcal{P}'$ (Definition 2.2) are considered \textit{similar} **if they differ only in a hyperparameter setting i.e.,** there is some $j \in [m]$ for which $\lambda_k = \lambda_k’$ except when $j = k$ and $|\lambda_j - \lambda_j’| \leq \tau_j$.”
> > >
> > > This now directly aligns with the consistency experiments that we run in our benchmark, where we fix randomness and vary a single hyperparameter across runs (thus, all results for each of our consistency plots compare similar learning pipelines according to this definition). Thanks again!
> > >
> > > > **Def of consistency as fairness axiom, relationship to fairness [i.e. comment “...b) it's super fairness related …”]**
> > >
> > > Your point is well taken. We likely won’t do much more to convince you that consistency is an important fairness consideration, besides to frame the following counterfactual: for two similar learning pipelines that produce nearly identical predictions on the same task, suppose their associated uncertainty estimates differ wildly due solely to a minor change in a single hyperparameter - perhaps the random seed for model initialization, or changing a regularization parameter from 0.5 to 0.6. Is it fair that such an insignificant difference leads to substantial disparities in uncertainty assessments? This variability means that individuals could receive markedly different uncertainty estimates purely because of an arbitrary hyperparameter choice, not because of any meaningful change in the data or model structure. We argue this is a fairness concern, but respect that you may view it as more of a model class stability property.
> > >
> > > > **Def of consistency as fairness axiom, invariance to hyperparameters [i.e. comment “..a) this definition achieves tha …”]**
> > >
> > > After further reviewing your comment, and our definition, we think that we could restate the consistency definition formally as follows, while still respecting the existing benchmark.
> > >
> > > **Consistency**  For any two learning pipelines $\mathcal{P}$ and $\mathcal{P}'$ with hyperparameters $\lambda$ and $\lambda'$, let $\delta(\lambda, \lambda')$ be a distance metric quantifying the difference between hyperparameters. The predictive functions $f_\mathcal{P}$ and $f_{\mathcal{P}'}$ produced by these pipelines should satisfy:
> > > $$
> > > \| \sigma - \sigma' \| \leq L \cdot \delta(\lambda, \lambda'),
> > > $$
> > > where $\sigma$ and $\sigma'$ are the uncertainty estimates from $f_\mathcal{P}$ and $f_{\mathcal{P}'}$, respectively, and $L$ is a Lipschitz constant.
> > >
> > > Would stating it in this manner (or a similar manner) help with your suggestion to state the definition along the lines of “bounded change in output for any change in input?” If you believe this is a clearer definition, and more aligned with fairness, we’d be happy to update the definition of consistency in our paper.
> > >
> > > > **Calibration metrics …  adding more literature to the related work (eg "Measuring Calibration in Deep Learning" is an example paper from this literature)**
> > >
> > > Thanks for calling our attention to this again; we appreciated your initial suggestion, as the ECE metric has strengthened the clarity of our calibration results as is! We are additionally working on putting together a discussion on the calibration literature, starting with the “Measuring Calibration in Deep Learning” paper, and papers they cite/who cite them. We will post a version of that discussion here when we are done, for you to review, and then add it to our paper body.
> > >
> > > **Thank you for continuing to engage with our work, and for the suggestions. Please let us know if we can make other changes that you feel would strengthen the paper.**

---

> > > > ### Comment · Reviewer_ufZ7 · 2024-11-26
> > > >
> > > > Responding below:
> > > >
> > > > Consistency as fairness axiom: you're right that we may have to agree to disagree here. There is some old fairness work that I've seen around Lipschitzness wrt input as a fairness property but that seems quite different from what you're looking at here. Re: your example, a change in hyperparameter which yields a change in output is not necessarily unfair - in fact, since hyperparameter choice has a global effect there is a chance that it treats all individuals identically. For instance, it's possible that a hyperparameter change uniformly reduces all uncertainty estimates by X% - I would not necessarily call this an unfair impact. Of course a uniform shift across the input space is unlikely but I think the point holds that the (un)fairness of the impact seems somewhat orthogonal to the size of the change (which could be 5 or 50%).
> > > >
> > > > New def of consistency: I think this looks fairly similar to me as far as my previous points are concerned.

---

> > > > > ### Author Response · Authors · 2024-11-26
> > > > >
> > > > > > **Consistency as fairness axiom: a change in hyperparameter which yields a change in output is not necessarily unfair …**
> > > > >
> > > > > Understood! We propose the following path forward, which should serve to both address the point you’re making and broaden the paper’s discussion on fairness (to acknowledge your example of how swings in the magnitude of uncertainty are not inherently unfair under one interpretation of what constitutes fairness, but are not accounted for under the stated definition of consistency). Perhaps we can both agree that fairness concerns do arise when such large swings based on small hyperparameter changes disproportionately affect uncertainty estimates for protected groups in the data. Thus, one goal would be to ensure that minor hyperparameter adjustments do not introduce significant disparities in uncertainty estimates across different subpopulations i.e. consistency **across subgroups.**
> > > > >
> > > > > We could formalize this as follows,
> > > > >
> > > > > **Consistency Across Subgroups.** Let $\mathcal{P}$ and $\mathcal{P}'$ be two similar learning pipelines differing only in hyperparameters $\lambda$ and $\lambda'$, with $\delta(\lambda, \lambda')$ measuring the change in hyperparameters. For all protected groups $a, a' \in \mathcal{A}$, the change in uncertainty estimates should not disproportionately affect one group over another. Specifically, we require: $$ | (\sigma_a - \sigma_a' ) - ( \sigma_{a'} - \sigma_{a'}' ) | \leq L \cdot \delta(\lambda, \lambda'), $$
> > > > > where, $\sigma_a = \mathbb{E} [ \sigma(x) ]$  is the average uncertainty estimate for group $a$ under pipeline $\mathcal{P}$, $\sigma_a' = \mathbb{E} [ \sigma'(x) ]$ is the average uncertainty estimate for group $a$ under pipeline $\mathcal{P}'$, $L$ is a Lipschitz constant, and $\delta(\lambda, \lambda')$ quantifies the change in hyperparameters (note the expectations are taken under ${x | A = a}$).
> > > > >
> > > > > We could include both definitions for completeness, and note the philosophical distinction in what constitutes *fair consistency,* and how this leads to a more general definition or a more specific one. Does this seem like a reasonable way to strengthen the paper and adjust our definition of consistency to account for the points you brought up?

---

> > > > > > ### Comment · Reviewer_ufZ7 · 2024-11-27
> > > > > >
> > > > > > Yeah this seems like a closer match to what I would expect - I think this paper either should be reframed away from fairness a bit, or towards a definition of this rough flavor. That said it is a fairly significant edit, both from a theory + experimental perspective.

---

> > > > > > > ### Author Response · Authors · 2024-11-27
> > > > > > >
> > > > > > > As you said, we may have to agree to disagree on which exact philosophical framing of fairness and consistency makes the most sense. We do, however, appreciate the push to consider this alternative notion of consistency we've jointly developed, and we'll add the explanation and alternative definition to the paper to enrich that discussion.
> > > > > > >
> > > > > > > As a last appeal, we'd like to point out that our results in **Figures 18 and 19** (which show differences between groups in terms of different abstention and error rates) could be easily modified under the definition of consistency between subgroups. We could plot the per group average uncertainty (y-axis) vs. varying the hyperparameters across different models (the *max_depth* and *reduction_thresholds* for XGBoost and the $\alpha$ weight decay regularization parameter for the neural model). The current results in **Figures 18 and 19** seem to suggest that the per-group consistency would differ across different parameter settings, which would be good empirical results to pair with the new *Consistency Across Subgroups* definition. Let us know what you think of that proposal, and thank you again for continuing to engage with us in this review process!

---

> > > > > > > > ### Comment · Reviewer_ufZ7 · 2024-12-02
> > > > > > > >
> > > > > > > > Sorry for the delay, was away for Thanksgiving. With a reframe of the paper to include this idea substantially in terms of that definition both theoretically and experimentally, and providing clarity around the separation of the two definitions and how they provide differing criteria towards robustness and fairness respectively, I'd be happy to raise my score. I'm not sure this is a reasonable ask in the review process though.

---

> ### Author Response · Authors · 2024-11-21
>
> > **for consistency, it seems like a non-robust choice to output a maximum std. Dev over individuals - I’m wondering if like a 90th percentile might be a better choice**
>
> W5. Good point! We will run the final version of our experiments with the 90th percentile of the standard deviation. For now, we note that the max std aligns with the fuller summary of standard deviations. For example, in Figure 2, Ensemble is the most consistent, closely followed by Binomial NLL, while Self-(in)consistency and Selective Ensemble are the least consistent.
>
> > **odd that objective function is normalized sum of multiple fairness metrics**
>
> W6/Q4. This is a helpful remark—thank you! **We have updated Table 3 in the revised paper PDF. Now, we have one set of algorithms whose abstention rate is optimized for statistical parity and another set of algorithms whose abstention rate is optimized for equalized odds. With these updates, we find the same results: While the abstention framework allows models to reduce their error rate, it does not magically reduce the imbalance in outcomes between demographic groups.
>
> > **Def 6.2 - it seems overly specific to me to assume that the model outputs are parametrizing a normal distribution. In particular, I don’t think this makes sense in the binary Y case…**
>
> W7/Q5. You are correct! UA-SP does not make sense as a fairness metric in the binary setting; it is specific to the regression setting. In the regression setting, we make the assumption that the heteroscedastic uncertainty is normally distributed. We could also state UA-SP for a more general distribution class with more parameters but, for simplicity, we chose to state the UA-SP definition in the context of the normal distribution. Note that this assumption may be more or less applicable depending on the uncertainty.
>
> We could have also stated UA-SP in a more general sense, given here:
>
> $\textbf{Uncertainty-Aware Statistical Parity} (\textit{UA-SP})~~~$ Consider a function $f: \mathcal{X} \times \mathcal{A} \to \mathbb{R}^2$ that estimates a mean $\mu$ and a standard deviation $\sigma$. The predictions induce a randomized function $\tilde{f}: \mathcal{X} \times \mathcal{A} \to \mathcal{Y}$ that samples $y$ from a probability distribution with mean $\mu$ and variance $\sigma^2$ and PDF $P(y; \mu, \sigma^2)$. Then $f$ satisfies uncertainty-aware statistical parity if, for all protected groups $a \in \mathcal{A}$ and outcomes $y \in \mathcal{Y}$,
> $$
> Pr(\tilde{f}(\mathbf{X}, A) \geq y \mid A = a) = Pr(\tilde{f}(\mathbf{X}, A) \geq y).
> $$
>
> If you would like, we would be happy to update our definition of UA-SP to be the more general form in the revised PDF, and then specify that for practical purposes when evaluating, we set $P(y; \mu, \sigma^2) = \mathcal{N}(y; \mu, \sigma^2)$.
>
> *Small point weaknesses:* We appreciate these notes! We have updated such that it says “a pipeline that always leads to the same predictions is consistent but not meaningful.” When you say “clustering,” we assume you speak of experiments like the one presented in Figure 3; we group individuals based on the percentile buckets they fall into over their uncertainty estimates, and will clarify this in the final version. Disparate impact in Table 3 (of the revised PDF) is defined according to the ratio given in Definition I.4 (the last page of the Appendix). We will clarify that the fairness defintions/metrics can be found there in the final version of our paper.

---

> > ### Author Response · Authors · 2024-11-25
> >
> > We realize that the end of the ICLR rebuttal phase is a particularly busy time! Still, if you have a chance, we’d love to hear your thoughts on whether our response has addressed your concerns, or if there’s anything else we can do to clarify.

---

> ### Author Response · Authors · 2024-12-02
>
> We hope that you had a restful (and delicious) holiday period, thank you for getting back to us!
>
> This makes sense, and we appreciate your renewed openness. We believe that the following is within the purview of the review process: the proposal above, which is to introduce the new consistency across subgroups definition after the original definition, clearly define the contrast, and offer the consistency experiments as suggested.
>
> As we cannot revise the paper PDF any longer, **we have added the new section (slightly abbreviated below), and note that most of this new section follows the existing introduction of *consistency* / fairness philosophy in the current paper body, and some of which will be deferred to experimental results section. We have also added experiments on the Binomial NLL method with XGBoost on all the datasets to the main paper body from the below figures, and deferred the remaining experiments (on the other uncertainty methods / neural model) to the Appendix.**
>
> ### Refining Consistency: Consistency Across Subgroups
> The concept of *consistency* (as defined in Axiom 2.4) posits that similar learning pipelines should produce similar uncertainty estimates. This axiom emphasizes that uncertainty estimates should be a function of the data rather than arbitrary artifacts of the learning pipeline. However, one might argue that this does not necessarily connote unfairness; a minor adjustment in a hyperparameter could uniformly reduce all uncertainty estimates by a certain percentage. Such a uniform change *might* be deemed acceptable; fairness concerns would then only arise if the change disproportionately affects uncertainty estimates for protected groups. Though this scenario may be unlikely, it is important to address explicitly when defining *consistency* in a fairness context. Thus, we propose the following as an alternative *consistency* definition: **Consistency Across Subgroups**, which formalizes the idea that minor hyperparameter adjustments should not introduce significant disparities in uncertainty estimates across different subpopulations.

---

> ### Author Response · Authors · 2024-12-02
>
> **Consistency Across Subgroups** (*Axiom 2.4*) Let $\mathcal{P}$ and $\mathcal{P}'$ be two similar learning pipelines differing only in hyperparameters $\lambda$ and $\lambda'$, with $\delta(\lambda, \lambda')$ measuring the change in hyperparameters. For all protected groups $a, a' \in \mathcal{A}$, the change in uncertainty estimates should not disproportionately affect one group over another. Specifically, we require: \begin{equation} \left| \left( \sigma_a - \sigma_a' \right) - \left( \sigma_{a'} - \sigma_{a'}' \right) \right| \leq L \cdot \delta(\lambda, \lambda'), \end{equation} where $\sigma_a = \mathbb{E}_{x \sim \mathcal{D}a} [ \sigma(x) ]$ is the average uncertainty estimate for group $a$ under pipeline $\mathcal{P}$, $\sigma_a' = \mathbb{E}{x \sim \mathcal{D}_a} [ \sigma'(x) ]$ is the average uncertainty estimate for group $a$ under pipeline $\mathcal{P}'$, $L$ is a Lipschitz constant, and $\delta(\lambda, \lambda')$ quantifies the change in hyperparameters.
>
> *Axiom 2.3* focuses on the stability of uncertainty estimates across similar learning pipelines, treating all individuals equally, without considering group membership. However, in fairness-sensitive contexts, we might want to explicitly monitor and control changes in the learning pipeline that affect different subgroups, especially protected groups defined by attributes such as race, gender, or age. This is the utility of the **Consistency Across Subgroups** definition (Axiom 2.4), which extends the notion of consistency by explicitly accounting for potential disparities between groups. It emphasizes that uncertainty estimates should not only be stable overall but also that any changes should impact all groups similarly.
>
> To assess the practical implications of **Consistency Across Subgroups** (*Axiom 2.4*), we conducted experiments analyzing how hyperparameter variations affect uncertainty estimates across different protected groups. We focused on models varying in key hyperparameters, such as the maximum depth in XGBoost and the weight decay regularization parameter $\alpha$ in neural networks.
>
> **Figures 20 and 21** illustrate the per-group average uncertainty estimates as hyperparameters are varied. Specifically, we plot the average uncertainty $\sigma_a$ for each protected group $a$ against different settings of the hyperparameters. The results demonstrate whether changes in hyperparameters lead to disproportionate shifts in uncertainty estimates between groups. The experimental results reveal that, in some cases, minor hyperparameter adjustments can lead to significant differences in uncertainty estimates between groups, violating the **Consistency Across Subgroups** (*Axiom 2.4*) criterion. For example, in **Figure 20**, we observe that increasing the maximum depth in XGBoost models disproportionately increases the uncertainty estimates for one protected group compared to the other. Similarly, although to a lesser degree, **Figure 21** showed across all the datasets that adjusting the weight decay in neural networks can have unequal effects on different groups' uncertainty estimates. These experiments underline the utility in adopting **Consistency Across Subgroups** (*Axiom 2.4*) in fairness contexts, as the definition acknowledges that fairness in uncertainty estimation can be both about individual stability (*Axiom 2.3*) and also about equitable treatment across different subpopulations, depending the predictive setting and training pipeline being audited.

---

> > ### Author Response · Authors · 2024-12-03
> >
> > Oh shoot, it appears as though you are unable to add a new response today, even though the review period is still open. However, you should still be able to update your score if you choose to do so. Were you to raise your score, we would take that as a signal that you feel positively about our proposed changes in response to your suggestions. Thank you again for your time and review!

---

### Official Review · Reviewer_5aQp · 2024-11-02

**Soundness:** 3
**Presentation:** 3
**Contribution:** 3
**Rating:** 6
**Confidence:** 3

**Summary:**

The authors introduce FairlyUncertain, an axiomatic benchmark for evaluating uncertainty estimates in fairness. The benchmark posits that fair predictive uncertainty estimates should be consistent across learning pipelines and calibrated to observed randomness. FairlyUncertain suggests that: In the binary setting, natural uncertainty estimates beat complex ensemble-based approaches and abstaining improves error but not imbalance between demographic groups. In the regression setting, consistent and calibrated uncertainty methods can reduce distributional imbalance without any explicit fairness intervention.

**Strengths:**

1. The organization and writing are fairly clear, and the structure of the paper is sound.

2. The motivation is clear, and the background as well as the related works are well explained.

3. The paper shows enough originality and novelty, by providing a novel framework for fairness and uncertainty.

4. The experiments are extensive, comparing the performances on various datasets and different prediction tasks.

**Weaknesses:**

1. The connection between "Fairness" and "Consistency/Calibration" is not so clear: in the description of consistency/calibration metrics, the sensitive attributes don't seem to be included and discussed. For example, for consistency, all the metrics (SD or p-values) are computed over different ensembles, but how is it related to fairness? For calibration, it says "We identify groups of individuals with similar uncertainty estimates and empirically evaluate the standard deviation of the residual difference between the observed and predicted outcomes", but in the previous definition of calibration (Axiom 2.4), the group should be defined by sensitive attributes. The relationships between fairness and consistency/calibration should be further explained.

2. The methodology is limited to ensemble methods: the consistency/calibration metrics discussed in the paper all focus on ensemble methods (i.e., XGBoost), how can this be generalized to a wider class of ML methods (e.g., neural networks with stochastic optimizers)?

3. The experiments on fairness interventions are not very convincing: in section 5, the authors show that "abstaining from binary predictions, even with improved uncertainty estimates, reduces error but does not alleviate outcome imbalances between demographic groups", can the authors provide more insights into this phenomenon? This point is linked to my previous request for more explanation on the connection between fairness and uncertainty estimates.

**Questions:**

1. Is there more discussion on the connection between fairness and consistency/calibration? For example, how does the standard deviation across different hyper-parameters (which don't take sensitive attributes into account) help mitigate bias towards certain groups (which need information on sensitive attributes)?

2. Can this definition of consistency/calibration be generalized to other ML methods other than ensembles?

I would like to raise my score if these weaknesses/questions can be addressed.

---

> ### Author Response · Authors · 2024-11-21
>
> Thank you for taking the time to review our submission and for offering suggestions for improvement! We will respond to each of your comments/questions below, briefly summarizing the comment inline.
>
>
> > **clarify connection between "Fairness" and "Consistency/Calibration"**
>
> W1: Thank you for this insight; we note that similar concerns were raised by other reviewers, and we have addressed them by adding a new section titled **"Connecting Consistency and Calibration to Fairness Principles"** in our revised PDF.
>
> To clarify briefly, the connection between fairness and our axioms of consistency and calibration lies in how uncertainty estimates impact equitable decision-making. While our consistency metrics (e.g., standard deviation or p-values) are computed over different ensembles without explicit reference to sensitive attributes, they relate to fairness by mitigating arbitrary variability in predictions that could disproportionately affect sensitive groups. Ensuring consistent uncertainty estimates across models helps prevent unfair treatment arising from randomness in the learning process.
>
> Regarding calibration, we acknowledge that Axiom 2.4 involves groups defined by sensitive attributes. In our empirical evaluation, we grouped individuals with similar uncertainty estimates to assess calibration. However, we have clarified in the revised PDF that groups based on sensitive attributes are also essential for evaluating calibration in the context of fairness. Proper calibration ensures that uncertainty estimates are not systematically biased against any sensitive group, thereby promoting fair outcomes. Overall, we argue that consistency and calibration are crucial for fair uncertainty estimation because they help prevent arbitrary and biased predictions that could harm individuals (and furthermore exacerbate group disparities based on sensitive attributes).
>
> > **the methodology is limited to ensemble methods…how can this be generalized to a wider class of ML methods (e.g., neural networks)?**
>
> W2: We’d like to emphasize that our framework is not limited to ensemble methods. In fact, any model that can produce both a probability of a label (in the binary case) or a real value, is admissible. Still you bring up a valid point, also by other reviewers; to demonstrate that our framework is highly extensible to other methods, we have rerun experiments on neural networks with linear layers and non-linear ReLU activation functions (varying the $\alpha$ weight decay regularizer parameter). **You can see the results for this additional model class in **Figures 9 and 13** in the revised PDF.**

---

> ### Author Response · Authors · 2024-11-21
>
> > **can the authors provide more insights into how abstaining reduces error but does not alleviate outcome imbalances between demographic groups**
>
> W3. In response to this question, we have added the following illustrative example. Abstention is often employed as a fairness mechanism by utilizing uncertainty estimates, but our analysis shows that it does not inherently mitigate demographic disparities in binary classification tasks. In some scenarios, abstention can even cause or intensify biases. Consider a simple example involving a binary classification model evaluated on two demographic groups, $A$ and $B$, with $N_A = N_B$ examples. Both groups have the same number of positive ($Y = 1$) and negative ($Y = 0$) instances ($N_{A,1} = N_{A,0} = N_{B,1} = N_{B,0}$). Without abstention, the model performs flawlessly, achieving true positive rates (TPRs) and false positive rates (FPRs) of $\text{TPR}_A = \text{TPR}_B = 1.0$ and $\text{FPR}_A = \text{FPR}_B = 0.0$, satisfying the equalized odds criterion. Now suppose the model incorporates abstention based on uncertainty. For group $A$, the model exhibits extremely low uncertainty and continues to predict perfectly ($\text{TPR}_A = 1.0, \text{FPR}_A = 0.0$). However, for group $B$, the model demonstrates high uncertainty, abstaining on all positive instances ($Y = 1$), which leads to $\text{TPR}_B = 0.0$, while still correctly classifying negatives ($\text{FPR}_B = 0.0$). This disparity in TPRs ($\text{TPR}_A - \text{TPR}_B = 1.0$) introduces a severe violation of the equalized odds fairness metric. Although this is a hypothetical extreme, similar patterns of imbalance in abstention rates across subgroups appear in our experimental results, summarized in Table 4. A detailed discussion of this example is provided in Appendix E of the updated PDF (see Example~E.1).
>
> **In addition, we have added **Figures 18 and 19 to Appendix E** in the revised version of the paper.** These plots show how error rate decreases with a higher abstention rate when observations with high levels of uncertainty (and poor quality predictions) are abstained on. The key insight is that abstaining is independent of the protected groups and, as we see on several datasets like German and ACS Public Coverage and several methods like Selective Ensemble, abstaining can be done for one protected group at a higher rate than the other protected group, leading to **disparate error rates across protected groups**.
>
> >  **more discussion on the connection between fairness and consistency/calibration…**
>
> Great point! We have added a discussion to Section 2 of the paper, and added a version of it below for easy access.
>
> Uncertainty estimation is widely recognized as a crucial aspect of transparent machine learning practices (Bhatt et al., 2021; Hendrickx et al., 2024). But how is it connected to the concept of fairness? Uncertainty arises from variability in training data and randomness in learning algorithms, leading to a distribution of possible models rather than a single deterministic one. Ignoring this distribution risks making arbitrary decisions, especially for individuals whose predictions might vary across modeling decisions or other sources of uncertainty. Such arbitrariness could disproportionately and unfairly affect minority groups in data Tahir et al. (2023).
>
> Recent work has demonstrated that state-of-the-art fairness interventions can exacerbate predictive arbitrariness; models with similar fairness and accuracy performance but different parameters can assign vastly different predictions to individuals, and this arbitrariness is intensified by fairness constraints (Long et al., 2024; Cooper et al., 2024). Our axiom of consistency for fair uncertainty estimation builds upon this insight by asserting that uncertainty estimates should not vary significantly across similar learning pipelines. Furthermore, our axiom of calibration aims to prevent systematic biases in uncertainty estimates that could disadvantage certain groups. For instance, if uncertainty is consistently underestimated for a particular group, the model may overstate its confidence in predictions for that group, leading to unfair treatment (Ali et al., 2021). This leads us to argue that adhering to the axioms of consistency and calibration are necessary tenets of a fair uncertainty estimation process
>
> > **can this definition of consistency/calibration be generalized to other ML methods?**
>
> Q2. Certainly! There is nothing specific to ensembles about the method. As such, we defined consistency and calibration generally so they could apply to other machine learning models. For our experiments, we used XGBoost because it was more performant. At your request, we have also rerun experiments with a neural network, as mentioned above; overall, we find that the neural network is slower than XGBoost and produces very similar results.

---

> > ### Comment · Reviewer_5aQp · 2024-11-22
> > **Re: Official Comment by Authors**
> >
> > Thank you for your replies. I feel positive about your revision and I am willing to raise my score by 1.

---

> > > ### Author Response · Authors · 2024-11-24
> > >
> > > Thank you for taking the time to consider our revisions and rebuttal, and for raising your score! We’re available to address any remaining questions or concerns at your convenience before the deadline.

---

### Official Review · Reviewer_m3oY · 2024-11-03

**Soundness:** 3
**Presentation:** 3
**Contribution:** 3
**Rating:** 6
**Confidence:** 3

**Summary:**

The paper introduces "FairlyUncertain", a benchmark to evaluate predictive uncertainty in fairness contexts. It emphasizes the need for consistency and calibration of uncertainty estimates across learning pipelines. Experiments on ten datasets reveal that a simple uncertainty estimation method outperforms prior work, that abstaining from uncertain binary predictions reduces errors but not demographic imbalances, and that incorporating calibrated uncertainty in regression improves fairness. The benchmark is extensible and aims to standardize fairness assessments involving uncertainty, promoting equitable and trustworthy machine learning practices.

**Strengths:**

- Originality: Introduces FairlyUncertain, a novel benchmark for integrating uncertainty into fairness.

- Quality: Validated through extensive experiments across ten datasets; open-source and extensible.

- Clarity: Clear motivation, methodology, and presentation.

- Significance: Provides a foundational, scalable framework for enhancing fairness in machine learning.

**Weaknesses:**

- Findings on reducing errors but not outcome imbalances could be expanded with deeper insights or additional benchmarks.

- The paper does not fully explore variations in models, parameters, or datasets, which limits its generalizability and insights into broader use cases.

**Questions:**

- Are there plans to include additional fairness interventions?
- Which new calibration metrics could enhance FairlyUncertain?
- Why does regression improve fairness without explicit interventions?
- How generalizable are the results beyond the ten datasets used?

---

> ### Author Response · Authors · 2024-11-21
>
> Thank you for investing your time in reviewing our submission and for your constructive comments! We will respond to each of your comments/questions below, briefly summarizing the comment inline.
>
>
> > **Findings on reducing errors but not outcome imbalances could be expanded**
>
> W1: We have expanded our results to include experiments on per-group error rates in the abstention model. In Appendix E of the revised paper, Figures 18 and 19 plot the error rate for each protected group versus abstention rate. In Figure 18, we specify an overall abstention rate $r$ (on the $x$-axis) and plot the error rate on each protected group. We find that all the algorithms produce different levels of error between protected groups on the German and ACS Public Coverage datasets. In Figure 19, we plot the error rate (on the $x$-axis) and the abstention rate for each protected group. As we expect, the random baseline has no difference in abstention rate between protected groups (the uncertainty estimates are random). In contrast, Selective Ensemble has a substantial difference in abstention rates. This experiment has added depth to the benchmark, and we will continue adding similar experiments as FairlyUncertain grows! Please let us know if you have suggestions for another experiment.
>
> In addition, your point about reducing errors but not outcome imbalances has been echoed by other reviewers. When abstention is employed as a fairness strategy by utilizing uncertainty estimates, our results reveal that it does not necessarily reduce demographic disparities in binary classification tasks. In fact, abstention can sometimes create or amplify biases. To illustrate this with a straightforward example, consider a binary classification model evaluated on two demographic groups, $A$ and $B$, where each group contains $N_A = N_B$ examples. Both groups have an equal distribution of positive ($Y = 1$) and negative ($Y = 0$) examples ($N_{A,1} = N_{A,0} = N_{B,1} = N_{B,0}$). Without abstention, the model performs perfectly, achieving true positive rates (TPRs) and false positive rates (FPRs) of $\text{TPR}_A = \text{TPR}_B = 1.0$ and $\text{FPR}_A = \text{FPR}_B = 0.0$, thereby satisfying equalized odds. Now suppose the model incorporates abstention based on uncertainty. For group $A$, the model maintains very low uncertainty and continues to predict perfectly ($\text{TPR}_A = 1.0, \text{FPR}_A = 0.0$). In contrast, the model exhibits high uncertainty for group $B$, leading it to abstain on all positive examples ($Y = 1$), resulting in $\text{TPR}_B = 0.0$, while still correctly classifying negatives ($\text{FPR}_B = 0.0$). This abstention-driven gap in TPRs ($\text{TPR}_A - \text{TPR}_B = 1.0$) introduces a significant violation of the equalized odds fairness metric. While this is an extreme case, we observe similar subgroup disparities in abstention rates in our empirical findings, detailed in Table 3. This example is further explained in Appendix E of the revised PDF (refer to **Example E.1**).
>
> > **fully explore variations in models, parameters, or datasets**
>
> W2: Our focus was on constructing a novel set of evaluations according to the principles of consistency and calibration, and on running across a variety of datasets and for a variety of metrics. We agree that including more models, parameters, etc. would strengthen the robustness of the results. To that end, we have included results for a neural network (linear layers with non-linear activations, see **Figures 9 and 13** in the revised PDF)
>
> Our benchmark is also highly extensible and we will continue to update with new model architectures, hyperparameter variations, and relevant datasets.
>
> > **plans to include additional fairness interventions**
>
> Q1: Yes, we have plans to include an extensive array of fairness intervention techniques. To demonstrate the robustness of our results **in Table 3,** we have added results on the state-of-the-art FairGBM in-processing fairness method (Cruz et. al 2022, 2024) for the revised rebuttal PDF. This is a variant of the strong LightGBM predictor (Ke et al, 2017), but with fairness constraints on the objective. We have two variants, FairGBM SP and FairGBM EO, each tuned for those respective fairness metrics. Note that this method performs well on fairness metrics, and has a lower error rate than some of the other fairness interventions, but still cannot match the error rate performance of the methods allowed to abstain (for the same reasons we discuss in Section 5).

---

> ### Author Response · Authors · 2024-11-21
>
> > **new calibration metrics to enhance FairlyUncertain?**
>
> Q2: This is a very good question. We default to negative log likelihood as it is very commonly used in loss function and has strong connection to information entropy. However, we have added results on another common calibration metric, Expected Calibration Error (ECE, Naeini et. al 2015). We also have plans to include other scoring rules (Spherical, Brier, etc.). We will discuss ECE in comparison to NLL here (and note this section has also been added in our revised PDF, in the appendix).
>
> Expected Calibration Error (ECE) and Negative Log Likelihood (NLL) will differ fundamentally in their model calibration assessment. Given predictions $[p_i]^N_{i=1}$, uncertainty estimates $[\sigma_i]^N_{i=1}$, and true labels $[y_i]^N_{i=1}$, ECE groups predictions into $M$ confidence bins $[B_m]^M_{m=1}$ based on $p_i$, and computes calibration as
> $\text{ECE} = \sum_{m=1}^M \frac{|B_m|}{N} \left| \text{acc}(B_m) - \text{conf}(B_m) \right|$, where $acc(B_m) = \frac{1}{|B_m|} \sum_{i \in B_m}$ $1$ $( \hat{y}_i = y_i )$, while $conf(B_m)$ is the sum of each $p_i$ normalized by $\frac{1}{|B_m|}$.
>
> In contrast, our modified NLL incorporates uncertainty estimates directly by adjusting predicted probabilities as $\tilde{p}_i = (p_i > 0.5) p_a + (p_i \leq 0.5) p_b$, where $p_a = (1 + \sqrt{1 - 4 \sigma_i^2})/2$ and $p_b = (1 - \sqrt{1 - 4 \sigma_i^2})/2$. We then compute NLL in standard fashion e.g. $NLL = -\frac{1}{N} \sum^N [ y_i \log(\tilde{p}_i) + (1-y_i) \log(1-\tilde{p}_i) ].$
>
> ECE provides an interpretable, aggregate view of calibration by measuring the alignment of predicted probabilities with empirical accuracy in confidence intervals (Naeini et al, 2015). However, it is sensitive to binning choices and lacks granularity at the individual prediction level. Our adjusted NLL method avoids binning, directly incorporating uncertainty estimates to evaluate calibration at a finer resolution, penalizing overconfident errors and underconfident correct predictions. While this makes NLL more sensitive to uncertainty quality, it may conflate calibration with model discrimination, and its dependence on predicted standard deviations assumes valid uncertainty estimates. We'd expect to prefer something like ECE for global calibration trends, while our NLL-based approach is suited to uncertainty-aware evaluation at the individual level. This makes the metrics complementary, and we thank the reviewer for the suggestion of adding additional calibration metrics to the benchmark, to better capture distinct aspects of calibration performance. **Comments and explanations for ECE have been added to Appendix Section D.1.**
>
>
> > **Why does regression improve fairness without explicit interventions**
>
> Q3: Thank you for highlighting this result, we are happy to offer some thoughts on the matter here. The \textit{Normal NLL} method likely improves fairness without explicit interventions due to its ability to account for heteroscedasticity in the data. Its loss function, $-\log \sigma + \frac{1}{2}\left(\frac{y - \mu}{\sigma}\right)^2,$ includes terms that regulate the variance ($\sigma$), preventing it from becoming arbitrarily large while encouraging larger $\sigma$ for high residuals ($|y - \mu|$). This adaptive mechanism ensures accurate estimation of variance for each prediction. By modeling variance as a function of input features, which may include or correlate with the protected attribute $A$, the method captures group-specific heteroscedasticity.
>
> When predictions include uncertainty (e.g. assumed drawn from $y \sim \mathcal{N}(\mu, \sigma^2)$), the resulting cumulative distribution functions (CDFs) are smoothed, particularly for high-uncertainty predictions. This smoothing reduces sharp differences between groups, aligning the predictive distributions across protected groups. As \textit{Normal NLL} is both consistent and calibrated, its uncertainty estimates reliably reflect true uncertainties, minimizing group-specific biases. This supports fairness by enabling the model to satisfy the uncertainty-aware statistical parity (\textit{UA-SP}) condition: $\Pr(\tilde{f}(\mathbf{X}, A) \geq y \mid A = a) = \Pr(\tilde{f}(\mathbf{X}, A) \geq y),$ where $\tilde{f}$ incorporates group-specific uncertainties. To summarize, by smoothing predictions through accurate variance estimation, the method naturally reduces disparities across groups, and thus this likely explains why we observe some fairness improvements.

---

> ### Author Response · Authors · 2024-11-21
>
> > **generalizability of results**
>
> Q4: This is a good question! We specifically chose a reasonably large, representative sample of commonly available fairness benchmark datasets (including general support for the ACS folktables package (Ding et. al 2021, https://github.com/socialfoundations/folktables) which itself contains many different data scenarios). We believe we have already shown pretty extensive results in the tabular data setting, and are excited to continue expanding and updating the available datasets to demonstrate the robustness of our findings.

---

> > ### Comment · Reviewer_m3oY · 2024-11-23
> >
> > Authors, thank you for your replies. After reading the comments, I do believe the paper can add value to the conference and would like to maintain my score

---

> > > ### Author Response · Authors · 2024-11-24
> > >
> > > Thank you for the additional time and effort spent on our revised submission and rebuttal! If you have any further questions or suggestions, we’d be happy to address them before the deadline.

---

### Official Review · Reviewer_LGgY · 2024-11-04

**Soundness:** 3
**Presentation:** 3
**Contribution:** 2
**Rating:** 6
**Confidence:** 3

**Summary:**

The paper introduces FairlyUncertain, a benchmark designed to assess the integration of uncertainty into fairness contexts within machine learning. Addressing the inherent prediction uncertainty challenge for fairness, the paper emphasizes that uncertainty estimates should ideally be consistent across similar models and calibrated to observed randomness. Through experiments across 10 datasets, including binary classification and regression tasks, the authors evaluate various methods for uncertainty estimation and introduce a new fairness metric, Uncertainty-Aware Statistical Parity (UA-SP), tailored for regression tasks. FairlyUncertain provides a structured benchmark to explore the nuanced relationship between uncertainty and fairness, specifically examining the effects of abstention and confidence thresholds.

**Strengths:**

1- FairlyUncertain introduces a standardized benchmark that is both theoretically grounded and practical, which allows researchers to evaluate how uncertainty affects fairness and vice versa. This is timely, given the increasing need for fair AI applications under uncertainty.

2- The paper defines fairness-focused axioms: Consistency (Axiom 2.3) and Calibration (Axiom 2.4), which set criteria for reliable uncertainty estimates. These axioms are clear and can be used for the practical goal of having fair predictive models that remain robust even under model variations.

3- The authors provide extensive evaluations of consistency and calibration across datasets and methods. They cover both abstention and confidence thresholding. The benchmark’s focus on these uncertainty handling methods, without endorsing one as definitive, offers a structured way to assess their impact on fairness.

4- The benchmark is open-source which could be used in future research.

5- The benchmark’s use of consistency as a measure for similar pipelines indirectly addresses concerns about model multiplicity. FairlyUncertain's evaluation across slight hyperparameter changes serves as a practical check for this multiplicity effect.

**Weaknesses:**

1- Axiom 2.3 suggests that small changes in hyperparameters should not significantly impact uncertainty estimates. While this is a reasonable assumption, it would benefit from explicit clarification on the stability limits of uncertainty under various hyperparameter changes to avoid overgeneralization.

2- Although the benchmark evaluates abstention as a fairness strategy, abstention does not reduce demographic disparities in binary classification tasks. Addressing why abstention fails to impact fairness in this context could inform potential adjustments to the benchmark. Also, given that abstention was shown not to reduce demographic disparities, the paper could address situations where abstention might inadvertently introduce biases. For example, explaining conditions under which abstention could be counterproductive would deepen the understanding of its impact on fairness.

3- While tables like Table 2 and Table 4 capture calibration and consistency scores, the paper could strengthen its analysis by discussing why certain methods perform better/worse across different datasets. Additionally, the appendix includes valuable demographic-specific breakdowns that could be better integrated into the main analysis to improve understanding of method reliability and demographic-specific impacts.

4- While XGBoost is effective for tabular data, expanding to deep learning models like MLPs, or transformers would align with recent trends in fairness research and test FairlyUncertain’s utility on more complex architectures. Reproducibility might vary across model architectures, especially complex ones, due to the implicit assumption of consistency under slight hyperparameter variations. While the benchmark is robust as presented, extending it to a broader model set might require tuning and validation.

**Questions:**

1- How does this work compare with existing research on fairness evaluation (beyond Section 6.1)? i.e., to what extent does the authors' focus on uncertainty help the practical selection of fairness measures and mitigation strategies?

2- Could this framework be adapted for more complex models or architectures and extended to additional fairness notions, such as intrinsic fairness? If so, how?

3- What are the societal implications of incorporating uncertainty in the selection of fair models? Could the authors expand on the strengths of their approach relative to existing methods, providing examples to illustrate the impact beyond simple numerical metrics? Existing literature on model multiplicity highlights the uncertainty involved and explores its societal, legal, and philosophical implications (there are many papers out there on this topic). I am wondering, what new insights does this paper contribute to these ongoing discussions?

---

> ### Author Response · Authors · 2024-11-21
>
> Thank you so much for the time you took to review our paper and provide constructive feedback! We will respond to each of your comments/questions below, briefly summarizing the comment inline.
>
> >  **Axiom 2.3 (consistency) … would benefit from explicit clarification on the stability limits of uncertainty under various hyperparameter changes**
>
> W1. Thank you for the suggestion to clarify stability limits for consistency; we have adjusted Axiom 2.3 to this end as follows:
>
> $\textbf{Consistency}~~$
> Let $\mathcal{P}$ and $\mathcal{P}'$ be similar learning pipelines as per \textbf{Definition 2.2}, differing only in hyperparameter $\lambda_j$ with $|\lambda_j - \lambda_j'| \leq \tau_j$. Let $f$ and $f'$ be the predictive functions produced by $\mathcal{P}$ and $\mathcal{P}'$, respectively. Then there exists a non-decreasing function $\delta_j: [0, \tau_j] \to \mathbb{R}_{\geq 0}$ with $\delta_j(0) = 0$, such that for all inputs $(\mathbf{x}, a) \in \mathcal{X} \times \mathcal{A}$, the uncertainty estimates satisfy,
> $$|\sigma(\mathbf{x}, a) - \sigma'(\mathbf{x}, a)| \leq \delta_j(|\lambda_j - \lambda_j'|).$$
> This means that small changes in hyperparameter $\lambda_j$ within the threshold $\tau_j$ lead to controlled variations in uncertainty estimates, bounded by $\delta_j$.
>
>
> >  **…addressing why abstention fails to impact fairness in this context could inform potential adjustments to the benchmark…explain conditions under which abstention could be counterproductive [to] deepen understanding of its impact on fairness**
>
> W2. Thank you for this comment, we have included a more in-depth discussion of abstention in the revised PDF, specifically addressing its impact on fairness metrics. While abstention is commonly used as a fairness strategy by leveraging uncertainty estimates, our findings demonstrate that it does not inherently reduce demographic disparities in binary classification tasks. In certain cases, abstention can inadvertently *introduce* or *exacerbate* biases.
>
> As a simple, illustrative example, consider a binary classification model evaluated on two demographic groups, $A$ and $B$, each with $N_A = N_B$ examples. Both groups contain an equal number of positive ($Y = 1$) and negative ($Y = 0$) examples ($N_{A,1} = N_{A,0} = N_{B,1} = N_{B,0}$). Without abstention, the model predicts perfectly, achieving a true positive rate (TPR) and false positive rate (FPR) of $\text{TPR}_A = \text{TPR}_B = 1.0$ and $\text{FPR}_A = \text{FPR}_B = 0.0$, satisfying equalized odds. Say the model incorporates abstention based on uncertainty, and for group $A$, the model has arbitrarily low uncertainty and continues predicting perfectly ($\text{TPR}_A = 1.0, \text{FPR}_A = 0.0$). However, the model has \textit{high uncertainty} for group $B$, and thus the model abstains on all positive examples ($Y = 1$), resulting in $\text{TPR}_B = 0.0$, while still predicting negatives correctly ($\text{FPR}_B = 0.0$). This abstention-induced disparity in TPRs ($\text{TPR}_A - \text{TPR}_B = 1.0$) violates the equalized odds fairness metric to an arbitrary degree. Though this an extreme example, we observe this *imbalance in subgroup abstention rates* in our experimental results, **highlighted in Table 3.** We have **included this example in Appendix E in the revised PDF (see Example E.1**).
>
> > **strengthen its analysis by discussing why certain methods perform better/worse across different datasets…demographic-specific breakdowns that could be better integrated into the main analysis to improve understanding of method reliability and demographic-specific impacts**
>
> W3. Thank you for highlighting these results. We have **extended our results to include experiments** examining per-group error rates in the abstention model. In **Appendix E of the revised paper, Figures 18 and 19** display the relationship between error rates and abstention rates for each protected group. In Figure 18, we set an overall abstention rate $r$ (shown on the $x$-axis) and plot the error rates for each protected group. Our results reveal that the algorithms yield varying error levels across protected groups on both the German and ACS Public Coverage datasets. In Figure 19, we present error rates (on the $x$-axis) alongside the abstention rates for each protected group. As expected, the random baseline exhibits no difference in abstention rates between protected groups, as its uncertainty estimates are random. In contrast, the Selective Ensemble demonstrates significant disparities in abstention rates across groups. This experiment adds valuable insight to our benchmark, and we plan to incorporate similar analyses as FairlyUncertain evolves. We welcome any suggestions for additional experiments!

---

> ### Author Response · Authors · 2024-11-21
>
> > **expanding to deep learning models like MLPs, or transformers would align with recent trends in fairness research and test FairlyUncertain’s utility on more complex architectures**
>
> W4. This is an excellent comment, one that was echoed by other reviewers! We have included a neural network (linear layers with non-linear ReLU activations) tuned for tabular data classification in our experiments, and **added results to the appendix of the paper (see **Figures 9 and 13** in the revised PDF).**
>
> We are excited to continue updating the benchmark with models from different model classes, and agree that expanding to recent transformer advances for tabular classification would be great! This is why we kept our evaluations general and our benchmark extensible.
>
> > **what extent does the authors' focus on uncertainty help the practical selection of fairness measures and mitigation strategies**
>
> Q1. Good question. Broadly, we believe an axiomatic approach is crucial to principally selecting fair algorithms in settings with unavoidable uncertainty. Our experiments demonstrate that algorithms designed to be “fair” vary widely in their ability to produce consistent and calibrated uncertainty measures. Practically, we find the Binomial NLL method to be the most calibrated and consistent method; as such, we would suggest at least using it as a baseline in practical experiments.
>
> > **be adapted for more complex models or architectures and extended to additional fairness notions, such as intrinsic fairness**
>
> Q2. Beyond the more complex models (e.g., the new neural network experiment described above), we would be happy to add additional fairness notions to complement the ones we already evaluate (e.g., statistical parity, equalized odds, equal opportunity, disparate impact, predictive parity, and difference in false positive rate). However, we’re unable to find a formal definition of “intrinsic fairness”, could you please point us in the direction of a work that defines this? Do you mean how fair the initial classification setup is, without fairness interventions? Thank you in advance for clarifying!
>
> > **societal implications of incorporating uncertainty in the selection of fair models… model multiplicity highlights the uncertainty involved [in prediction] and explores its societal, legal, and philosophical implications…”**
>
> Q3. You are correct: predictive uncertainty estimation is not merely a technical consideration but has profound implications for fairness and justice in algorithmic decision-making (Bhatt et, al 2021, Cooper et. al 2023). Our research contributes the following insight: naively incorporating uncertainty into fair models can lead to unpredictable and potentially adverse outcomes for certain demographic groups. For example, our empirical analyses demonstrate that while abstention methods - where models defer decisions under high uncertainty - can reduce overall error rates, they do not necessarily improve fairness metrics such as statistical parity or equalized odds. This unpredictability may exacerbate existing disparities and undermine trust in these systems. Furthermore, the arbitrary application of uncertainty estimates might violate anti-discrimination laws and regulations. Uncertainty estimation is also integral to procedural justice (Rawls et. al 1971), which concerns the fairness of the methods and procedures used to arrive at decisions. By advocating for uncertainty estimates that are consistent across similar models and calibrated to actual data variability, we provide a more robust foundation for ethical algorithmic decision-making. We have added this discussion to a section titled **Societal Implications in Appendix Section A of our revised PDF.**

---

> > ### Comment · Reviewer_LGgY · 2024-11-24
> > **Thank you**
> >
> > I'd like to thank the authors for their responses to all reviewers, I read their answers and would be happy to raise my score by 1.

---

> > > ### Author Response · Authors · 2024-11-24
> > >
> > > Thank you so much for the time you spent reading our revised paper and rebuttal and for improving your score! We'd be happy to respond to additional questions or comments as well.

---

### Official Review · Reviewer_JxsV · 2024-11-06

**Soundness:** 3
**Presentation:** 3
**Contribution:** 3
**Rating:** 5
**Confidence:** 5

**Summary:**

The paper introduces FairlyUncertain, a Python package designed to evaluate uncertainty in fairness for machine learning algorithms. The package proposes methods and standards to incorporate uncertainty estimation into fairness evaluations in predictive models.

**Strengths:**

1. The paper identifies and addresses the underexplored intersection of fairness and uncertainty in predictive modeling.
2. The study spans multiple datasets and predictive tasks, enhancing the benchmark's relevance and applicability.
3. Designing the benchmark as an open-source tool encourages ongoing development and community contributions.

**Weaknesses:**

1. The paper targets uncertainty as a critical factor in predictive fairness, given the real-world impact of machine learning decisions. However, it does not adequately justify the importance of the consistency and calibration axioms themselves within this context. Specifically, while uncertainty is broadly recognized as significant for transparent AI, the paper could strengthen its justification by explaining why consistency and calibration specifically matter for fair uncertainty estimation.

2. Although the paper targets "fair uncertainty," the fairness component feels less integrated with uncertainty until Section 5, where it discusses the abstention framework and Uncertainty-Aware Statistical Parity. The paper could benefit from introducing the importance of fairness alongside uncertainty earlier, so readers understand the relationship between these two aims from the outset.

3. This paper is largely an implementation paper, presenting a framework and accompanying package for uncertainty-fairness evaluation rather than introducing fundamentally new theoretical insights into either fairness or uncertainty estimation. Its main contribution lies in providing a package that can support standardized evaluations of uncertainty estimation methods within fairness contexts. Therefore, it may be better suited for a benchmark track or tools/package track than a general research track.

**Questions:**

None

---

> ### Author Response · Authors · 2024-11-21
>
> We appreciate the time you took to review our paper and suggest improvements! We will respond to each of your comments/questions below, briefly summarizing the comment inline.
>
> > **does not adequately justify the importance of the consistency and calibration axioms themselves within the context of predictive fairness**
>
> W1. We appreciate this comment greatly! After reviewing our paper body, we agree. To address this, we have included a short section in our revised PDF titled **“Connecting Consistency and Calibration to Fairness Principles.”** We will restate the content here, written in a slightly more conversational style, for convenience.
>
> Uncertainty estimation is widely recognized as a crucial aspect of transparent machine learning practices (Bhatt et. al 2021, etc.). So how does this connect to fairness? Uncertainty arises due to variability in training data and randomness in learning algorithms (as we discuss in the intro, outline in Figure 1, Table 1, etc.), leading to a distribution of possible models rather than a single deterministic one. Ignoring this distribution can result in arbitrary decisions, especially for individuals whose predictions might vary across different modeling choices or sources of uncertainty. Such arbitrariness could disproportionately and unfairly affect minority groups in the data (e.g. affect fairness in a way that would be unaccounted for if only considering fairness measures related to the prediction).
>
> Recent work has shown that state-of-the-art fairness interventions can actually exacerbate this kind of “predictive arbitrariness” (Long et. al 2024, for example). Models that are similar in terms of fairness and accuracy but have different parameters can assign vastly different predictions to individuals, and fairness constraints can intensify this issue. Our axiom of consistency for fair uncertainty estimation builds on this insight by asserting that uncertainty estimates should not vary significantly across similar learning pipelines. But consistently bad uncertainty estimation or subgroups in the data could satisfy this axiom while potentially failing to achieve fairness. Our axiom of calibration aims to prevent this case; a calibrated uncertainty estimate would avoid systematic biases in uncertainty estimates that could disadvantage certain groups. If, conversely, a model consistently underestimated uncertainty for a particular group, it could overstate its confidence in predictions for that group, leading to unfair treatment. Therefore, we argue that adhering to the axioms of consistency and calibration is a necessary component of any fair uncertainty estimation process.
>
> > **benefit from introducing the importance of fairness alongside uncertainty earlier, so readers understand the relationship between these two aims from the outset**
>
> W2. Again, we appreciate your constructive observation regarding clearer integration of fairness and uncertainty. As we noted in our response to (W1) above, we have included a section early in the paper that directly addresses this comment. To summarize, we argue that **consistency** ensures that uncertainty estimates are stable across similar models, preventing arbitrary disparities in predictions. **Calibration** ensures that uncertainty estimates are accurate across different groups, preventing systematic biases that could disadvantage certain groups.
>
> > **better suited for a benchmark track or tools/package track than a general research track**
>
> W3. Thank you for your feedback on the positioning of our contribution. While our paper does present a framework and accompanying package, we believe it offers more than an implementation - it introduces non-trivial experimental setups and metrics for evaluating uncertainty and fairness and provides insights into the results of extensive evaluations. Additionally, to your point about conference suitability, ICLR includes "datasets and benchmarks" among its Subject Areas (https://iclr.cc/Conferences/2025/CallForPapers), and there have a number of influential benchmarks published at ICLR on fairness related themes in the past couple of years (Han et. al 2024, https://openreview.net/pdf?id=TzAJbTClAz , Cruz et. al 2024 https://openreview.net/pdf?id=jr03SfWsBS , Zong et. al 2023 https://openreview.net/pdf?id=6ve2CkeQe5S , to name a few.)

---

> > ### Author Response · Authors · 2024-11-25
> >
> > We are mindful of how hectic the end of ICLR rebuttal period can be! Still, if possible, we’d be grateful for your thoughts on whether our response has adequately addressed your feedback (and, of course, if you have any further comments/questions).

---

> > > ### Comment · Reviewer_JxsV · 2024-11-26
> > > **Reply to authors' feedback**
> > >
> > > Thanks for the reply. I am glad to see my comments have been adopted. Additionally, I reviewed the links to the ICLR Call for Papers and some past papers. I agree that this paper is within the scope.
> > >
> > > I acknowledge the valuable contribution of this paper in developing a benchmark across multiple datasets focused on fairness and uncertainty. However, I have reservations about the contribution of this paper integrating uncertainty evaluation into fair machine learning and the overall impact of this integration on this field. I am inclined to maintain my current rating. Thanks.

---

### Author Response · Authors · 2024-11-21
**Revised PDF**

We thank all of the reviewers for their time and insightful feedback! We’d like to highlight that we have uploaded a revised PDF, which includes any additional experimental results run as part of our rebuttal, alongside some lightly modified text. All additional changes as part of the rebuttal are highlighted as **blue text** (for new figures/tables for the rebuttal, the captions are in blue). **Then, in our rebuttal text below, we reference figures and tables according to the numbering in this revised, rebuttal PDF.**

---

### Author Response · Authors · 2024-12-03

We thank all reviewers for their time and effort in reviewing our paper. Below, we provide a global overview of the updates we made during the rebuttal process:

1. **Clarified the connection between fairness and our axioms of consistency and calibration**. We added a new section titled "Connecting Consistency and Calibration to Fairness Principles" to strengthen the philosophical foundation of our approach.
2. **Expanded our experimental results.** We added new experiments on per-group error rates and the impact of abstention on fairness metrics. We also extended our existing experiments with additional architectures, calibration metrics, and fairness definitions.
3. **Refined our definitions and formalizations.** Based on reviewer feedback, we enhanced the clarity and completeness of our presentation, particularly in the definitions of consistency and the formalization of our fairness/uncertainty axioms.

Once again, thank you very much to the area chair and all reviewers!

---

### Meta-Review · Area_Chair_bxVW · 2024-12-20

**Metareview:**

This paper introduces FairlyUncertain, a tool that evaluate uncertainty in fairness for machine learning algorithms. This framework emphasizes consistent uncertainty estimation and be calibrated to observed randomness.

The paper falls a bit short in its contribution. The main observations and reported findings suffer from generalizability without further exploring variations in models, datasets and fairness measures. In addition, the paper can benefit from better justifying the choice of computing the consistency and calibration of models.

**Additional Comments On Reviewer Discussion:**

After rebuttal, the reviewers had some remaining concerns that made them hesitate to recommend an acceptance, including the technical contribution being a little bit marginal.

---

### Decision · Program_Chairs · 2025-01-22

Reject